# Lattice engineering enables definition of molecular features allowing for potent small-molecule inhibition of HIV-1 entry

Yen-Ting Lai [1], Tao Wang[2], Sijy O'Dell[1], Mark K. Louder [1], Arne Schön[3], Crystal S. F. Cheung[1], Gwo-Yu Chuang[1], Aliaksandr Druz[1], Bob Lin[1], Krisha McKee[1], Dongjun Peng[1], Yongping Yang[1], Baoshan Zhang[1], Alon Herschhorn[4,5], Joseph Sodroski[4], Robert T. Bailer[1], Nicole A. Doria-Rose[1], John R. Mascola[1], David R. Langley[6] & Peter D. Kwong[1]

Diverse entry inhibitors targeting the gp120 subunit of the HIV-1 envelope (Env) trimer have been developed including BMS-626529, also called temsavir, a prodrug version of which is currently in phase III clinical trials. Here we report the characterization of a panel of small-molecule inhibitors including BMS-818251, which we show to be >10-fold more potent than temsavir on a cross-clade panel of 208-HIV-1 strains, as well as the engineering of a crystal lattice to enable structure determination of the interaction between these inhibitors and the HIV-1 Env trimer at higher resolution. By altering crystallization lattice chaperones, we identify a lattice with both improved diffraction and robust co-crystallization of HIV-1 Env trimers from different clades complexed to entry inhibitors with a range of binding affinities. The improved diffraction reveals BMS-818251 to utilize functional groups that interact with gp120 residues from the conserved β20-β21 hairpin to improve potency.

[1] Vaccine Research Center, NIAID, National Institutes of Health, Bethesda, MD 20892, USA. [2] Departments of Discovery Chemistry and Molecular Technologies, Bristol-Myers Squibb Research and Development, Cambridge, MA 02142, USA. [3] Department of Biology, Johns Hopkins University, Baltimore, MD 21218, USA. [4] Department of Cancer Immunology and Virology, Dana–Farber Cancer Institute, Harvard Medical School, Boston, MA 02215, USA. [5] Division of Infectious Diseases and International Medicine, Department of Medicine, University of Minnesota, Minneapolis, MN 55455, USA. [6] Computer Assisted Drug Design, Bristol-Myers Squibb Research and Development, Wallingford, CT 06492, USA. [7] Present address: Platform Chemistry, Arvinas Inc., New Haven, CT 06511, USA. Correspondence and requests for materials should be addressed to P.D.K. (email: pdkwong@nih.gov)

The entry of HIV-1 into target cells is a critical event in the viral life cycle and a target for drug development[1]. Viral entry is mediated by the HIV-1 envelope (Env) glycoprotein trimer, a type 1 fusion machine comprised of three gp120 subunits and three noncovalently linked gp41 subunits, which binds to cell-surface receptors and fuses viral and host cell membranes[2]. Entry inhibitors targeting the gp120 subunit have been developed[3], with a promising small-molecule lead, fostemsavir, the prodrug version of **BMS-626529** (temsavir) currently in phase III clinical trials[4]. Notwithstanding its advanced development and novel mode of action, next-generation inhibitors of fostemsavir have been sought to improve ADME (absorption, metabolism, distribution and elimination) profile[5], to overcome expected drug resistance[6], and to increase potency. We note that these qualities may be related; for example, increasing the potency of an inhibitor can be an effective way to counter drug resistance[7], as resistance mutations generally have only incremental effects on the dose−response of a drug[8].

X-ray crystallography is often instrumental in determining drug-binding mode and in facilitating structure-based drug design[9–11]. However, structure-based drug design can only be reliably carried out with crystals that diffract with resolution sufficient to provide accurate structural models; unfortunately, this resolution prerequisite has been difficult to achieve for many drug targets, even with extensive screening of crystallization conditions and protein variants[12]. Crystal engineering[13,14] represents an alternative strategy for crystal improvement, whereby inspection of a lattice with poor diffraction identifies weak lattice contacts, which can then be altered through structure-based design. However, both of these strategies can inadvertently introduce modifications that change the properties of protein targets and even their structures[15,16].

Crystallization chaperones, such as antibody fragments, have also been used to facilitate formation of crystal lattice contacts for difficult protein targets[17]. We recently reported the structure of **BMS-626529** (temsavir) in complex with an HIV-1 Env trimer bound by crystallization chaperones comprising the antigen-binding fragments (Fabs) of antibodies 35O22 and PGT122 (ref. [18]). We also reported the structure of **BMS-378806** (ref. [18]), the prototype small molecule for this class of compounds, in the same Env-35O22-PGT122 lattice. In both cases, the resolution was only 3.8 Å, and there was uncertainty in the positioning of small-molecule atoms and in the definition of side-chain interactions.

To obtain structural information of improved accuracy, we test a strategy involving the lattice-based engineering of crystallization chaperones. This strategy provides a way to improve a lattice without altering the protein target. We engineer crystallization chaperones to identify a crystal lattice suitable for determining high-resolution structures of inhibitors, spanning a range of >6000-fold neutralization potency, in complex with envelope trimers of clade A and B HIV-1 strains. We use this lattice to examine small-molecule inhibitors related to **BMS-626529** and report structures of multiple small-molecule inhibitors, including that of **BMS-818251**, an HIV-1 entry inhibitors with >10-fold higher potency than **BMS-626529**, which reveal structural determinants of potent HIV-1 inhibition and provide insights into the design of better entry inhibitors for this class of HIV-1 drugs.

## Results

**BMS-818251 shows >10-fold increased potency over temsavir.** By screening a library of temasvir derivatives, we identified two compounds, **BMS-814508** and **BMS-818251**, which showed improved entry inhibition of the laboratory-adapted HIV-1 strain NL4-3. The $EC_{50}$ for **BMS-814508** and **BMS-818251** was 0.495 ± 0.069 and 0.019 ± 0.003 nM, respectively, 4-fold and ~100-fold more potent than **BMS-626529**, which had an $EC_{50}$ of 2.2 ± 0.6 nM against the same strain[19]. Both of the improved compounds used a cyano alkene to replace an amide group with different thiazole substituents replacing the triazole on the 6-azaindole core of **BMS-626529** (temsavir) (Fig. 1a, Supplementary Fig. 1).

To verify neutralization potency seen in the neutralization-sensitive NL4-3 strain, we further tested the ability of these compounds to inhibit entry on a cross-clade panel of 30 HIV-1 strains derived from primary isolates (Fig. 1b). The 30 strains from all major HIV-1 clades were chosen to encompass the complete range of neutralization sensitivity towards **BMS-626529** in a 208-strain panel (ref. [18]). To provide comparative data, we assessed the well-studied entry inhibitors **BMS-378806** and **BMS-626529** as controls, and also included **BMS-386150**, which is very similar to **BMS-378806**, with a bromine modification in the compound to aid crystal structure determination (Fig. 1a). A recently reported compound **484**, which exhibits broad reactivity despite its moderate micro-molar range potency[20], was also tested. Of note, these six compounds all derive from a piperazine/piperidine bridge, with various degrees of modification, representing a collection of diverse entry inhibitors targeting the gp120 subunit of the HIV-1 Env trimer (Fig. 1a and Supplementary Fig. 1).

Neutralization of the 30-strain panel generally recapitulated prior observations (Fig. 1b, c). **BMS-626529** neutralized this panel of viruses with a geometric mean $IC_{50}$ of ~0.040 μM, which was within a factor of three of the geometric mean $IC_{50}$ of 0.015 μM that has been previously reported based on its inhibition of the 208-strain panel[18]. Consistent with the literature, **BMS-626529** (temasvir) was substantially more potent than **BMS-378806**, and compound **484** neutralized 11 of the 30 viruses with micro-molar potency. Despite an overall similarity with **BMS-378806**, **BMS-386150** had a substantially weaker (~7-fold lower) neutralization potency, and **BMS-814508** neutralized 28 of the 30 viruses with a mean inhibition similar to that of **BMS-626529** (temsavir). Cytotoxicity was observed for **BMS-814508** at the highest concentration (20 μM) in the neutralization assay (Supplementary Fig. 2). Strikingly, compound **BMS-818251** (geometric mean $IC_{50}$ ~ 0.002 μM) showed roughly 20-fold higher neutralization potency than **BMS-626529** (geometric mean $IC_{50}$ ~ 0.04 μM) on this 30-strain panel (Fig. 1b, c), with more than half of the viruses neutralized in the subnanomolar range.

The superior potency of **BMS-818251** observed on the 30-strain panel prompted us to assess the neutralization potency of this compound on the larger 208-strain panel (Supplementary Table 1). The geometric mean $IC_{50}$ of **BMS-818251** on the 208-strain panel was 0.0015 μM, in close agreement with the geometric mean $IC_{50}$ obtained from the 30-strain panel. When compared to the neutralization potency on the same 208-strain panel, **BMS-818251** has a 10.6-fold higher geometric mean $IC_{50}$ and 18-fold higher median $IC_{50}$ compared to **BMS-626529** (temsavir) (median $IC_{50}$s were 0.0005 and 0.0090 μM for **BMS-818251** and **BMS-626529**, respectively; geometric mean $IC_{50}$s were 0.0015 and 0.0159 μM, respectively).

To assess whether the improved potency of **BMS-818251** over temsavir was reflected by higher affinity to the HIV-1 Env trimer, we determined the binding affinity of **BMS-818251** to HIV-1 Env trimer by isothermal calorimetry (ITC). ITC showed **BMS-818251** to have an affinity of 0.047 μM to BG505 DS-SOSIP.664 Env trimer at 37 °C, which was substantially higher than the 0.402 μM affinity of **BMS-626529** under the same condition (Supplementary Fig. 3). Thus, the >10-fold improved potency of **BMS-818251** over temsavir in neutralization assays was associated with an 8.6-fold higher affinity, as determined by the ITC binding assay.

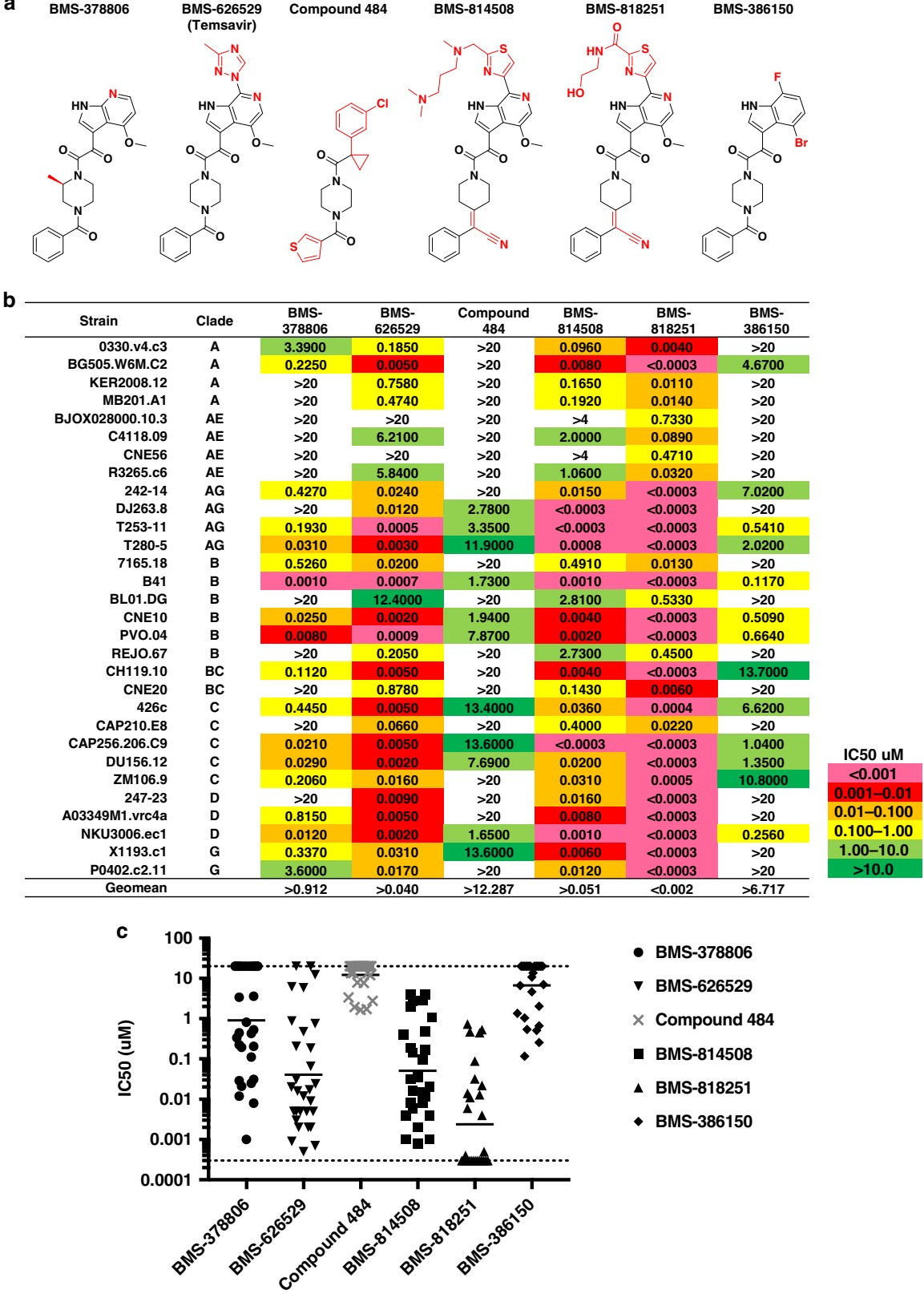

**Fig. 1** Diverse HIV-1 entry inhibitors span >6000-fold differences in neutralization potency, with **BMS-818251** being >20-fold more potent than BMS-626529 (temasvir). **a** HIV-1 entry inhibitors with common functional groups shown in black and unique features in red. **b** Neutralization assay of entry inhibitors against thirty HIV-1 isolates from all major HIV-1 clades. **c** Neutralization data shown as a scatter plot, with the geometric mean shown as horizontal bars. Dotted lines show the detection limits of neutralization assay

**Lattice engineering enables higher-resolution structures.** To provide a structure−function relationship between these small-molecule leads and their HIV-1 target, we sought to crystallize these compounds with the HIV-1 Env trimer. Previously we determined cocrystal structures of **BMS-626529** and **BMS-378806** at a resolution of 3.8 Å, bound to the BG505 SOSIP.664 trimer along with two crystallization chaperones, Fab 35O22 and PGT122 (ref. [18]). The BG505 SOSIP.664 Env trimer is a near-native Env trimer comprising fully cleaved gp120 and gp41 subunits, derived from strain BG505, with stabilized disulfide (SOS) and I559P (IP) mutation and truncated at residue 664 (ref. [21]). These cocrystal structures defined the general pose of drug compounds in the gp120 binding pocket, but left uncertain the atomic position of chemical substituents. To improve the resolution of the diffraction from this lattice (Fig. 2a, b), which allowed determination of several Env structures in independent studies (Supplementary Table 2), we attempted to reduce lattice flexibility in the crystallization chaperones (Fab 35O22 and Fab PGT122). The constant region of the 35O22 Fab appeared hyper-flexible as judged by its high B-factor (Supplementary Fig. 4a). Removal of the constant region by converting the Fab 35O22 to an scFv led to crystals that retained the original lattice, with almost identical lattice parameters (Supplementary Table 3). In structures of the 35O22 scFv, we observed some regions to be more flexible than in the Fab context, mainly due to a lack of structural support of antibody constant regions. To mitigate this flexibility, we tested two replacement variants: (i) replacing residues [10]ELK[12] with β-strand-preferring residues [10]TTT[12]; (ii) altering A16S and F84S to reduce exposed hydrophobic atoms; named 35O22_3T2S as in Supplementary Fig. 4b. Crystals generated with 35O22_3T2S has a substantially lower Wilson B factor of 28 Å$^2$ compared to 86 Å$^2$ for the original crystal (Fig. 2c). The refined structure of scFv 35O22_3T2S showed an average B-factor of 71 Å$^2$ vs. 202 Å$^2$ for the 35O22 Fab in the previous **BMS-626529** complexed structure. In addition to mutations introduced into 35O22, two methionine mutations (MM) were introduced onto the Fab of 3H109L, a precursor of PGT122 found to crystallize better[22]. These methionine mutations potentially interact with the base of gp41 molecules, one layer higher in the lattice (Fig. 2b, d). Finally, two potential glycosylation sites were introduced onto the 35O22 scFv, which enhanced its production yield by tenfold (Supplementary Fig. 4b).

To assess the impact of our lattice engineering, we determined the structure of **BMS-378806** in the improved lattice with the modified antibody fragments (termed 35O22_3T2S and 3H109L_MM). We observed almost identical unit cell dimensions as the original lattice[18], but with superior diffraction (Fig. 2c and Supplementary Table 3). Along the three principle directions $a^*$, $b^*$, and $c^*$, diffraction improved from 5.1, 4.8, and 3.0 Å to 3.3, 3.1, and 2.3 Å, respectively, as judged by the UCLA anisotropy server[23], with the unique reflections increasing from 22,236 to 51,309 (Fig. 2c). Because the space group (P6$_3$) remained the same and the unit cell dimensions were almost identical, the effective information content and data-to-parameter ratio obtained from the new crystal was more than double that of the parent lattice (PDB code: 4TVP) (Supplementary Table 3). The resulting electron density maps showed superior density for many regions of the Env trimer. For example, residues 65–73 in the constant region 1 (C1) of gp120 could now be accurately modeled with histidine residues at positions 66 and 72 forming an unusual stacking interaction, which was modeled differently in prior structures (Supplementary Fig. 5). Another region of improved structure modeling occurred at the α9 helix of gp41, where weak electron density previously made placement of amino acid side chains ambiguous[22,24,25]; the clear density in the new structure allowed unambiguous placement of amino acid side

chains and correct registry of backbone atoms (Supplementary Fig. 6). It is important to note that the C1 region and the α9 helix did not directly contact the small-molecule inhibitor; thus the improved resolution of these regions was a consequence of the improved lattice, not an effect of small-molecule stabilization. We also noted that for very dynamic regions of the Env (such as the inter-helical region of gp41), lattice engineering was unable to overcome the intrinsic flexibility of these regions, and they remained unresolved in our crystal structures. While the real space correlation for **BMS-378806** only improved from 0.91 to 0.95, the improved electron density did allow for greater structural certainty in placement of small-molecule atoms such as the methyl on the piperazine ring or the rotameric state of the oxoacetyl moiety. Overall, the engineered lattice allowed for substantially greater certainty in the positioning of both small-molecule compound and interacting Env-trimer side chains.

**Structural determination of compound 484-Env interactions.** As a test of the engineered lattice, we chose to examine compound **484**, which previously resisted cocrystallization, presumably due to its micro-molar range activity[20]. A binding pose of compound **484** had previously been proposed based on computational docking[20]. We used the engineered lattice to determine the cocrystal structure of **484** with the BG505 SOSIP.664 Env trimer and compared it to the docking simulation. The cocrystal structure (Fig. 3a) was similar to the overall pose of **484** in the docking study, though with notable differences. The thiophene ring essentially flipped 180° between cocrystal structure and docking pose, a difference supported both by the clear electron density defining **484** in the new lattice and by the strong scattering from sulfur, which served to identify this atom in the thiophene ring. Additionally, the piperazine ring rotated ~45° leading the acetyl-cyclopropane and chloro-benzene to shift 1.5 Å toward the thiophene ring (Fig. 3a).

Overall, **484** interacted mainly with three gp120 fragments: the C-terminus of α1 helix (residues 107–117; colored cyan in Fig. 3b); the C-terminal part of the CD4-binding loop and subsequent strands (residues 369–385; colored magenta in Fig. 3b); and the β20−β21 hairpin (residues 423–436; colored green in Fig. 3b). Hydrophobic interactions dominated the interface between **484** and the BG505 SOSIP.664 Env trimer (Fig. 3c), with the piperazine ring stacked between two tryptophan residues, W427 of the β20−β21 loop and W112 of the α1-helix. Two residues outside of the three major fragments, residues V255 and M475, also provided hydrophobic interaction with **484**. The cocrystal structure of compound **484** thus demonstrated the ability of the engineered lattice to provide atomic-level detail on the interaction between a low potency, low affinity compound, and the HIV-1 Env trimer.

**Tail region of BMS-818251 is critical for potent inhibition.** We next investigated whether crystal structures derived from the engineered lattice could provide a structure−function explanation for the very high potency of **BMS-818251** against HIV-1 BG505 compared to the closely related **BMS-814508** (IC$_{50}$ values < 0.0003 vs. 0.008 μM for **BMS-818251** and **BMS-814508**, respectively, against BG505; Fig. 1b). These analogs differ only in the chemistry of their respective tail functional groups, the moieties extending from the thiazole ring (Fig. 1a). Examination of cocrystal structures of **BMS-818251** and **BMS-814508** with the BG505 SOSIP.664 Env trimer revealed the tail of **BMS-818251** to make hydrophilic interactions with BG505 trimer (Fig. 4a), while the tail of **BMS-814508** did not make defined contacts with BG505 trimer (Fig. 4b). Specifically, the tail of **BMS-818251** interacts with the side chains of K117, R429 and Q432 of

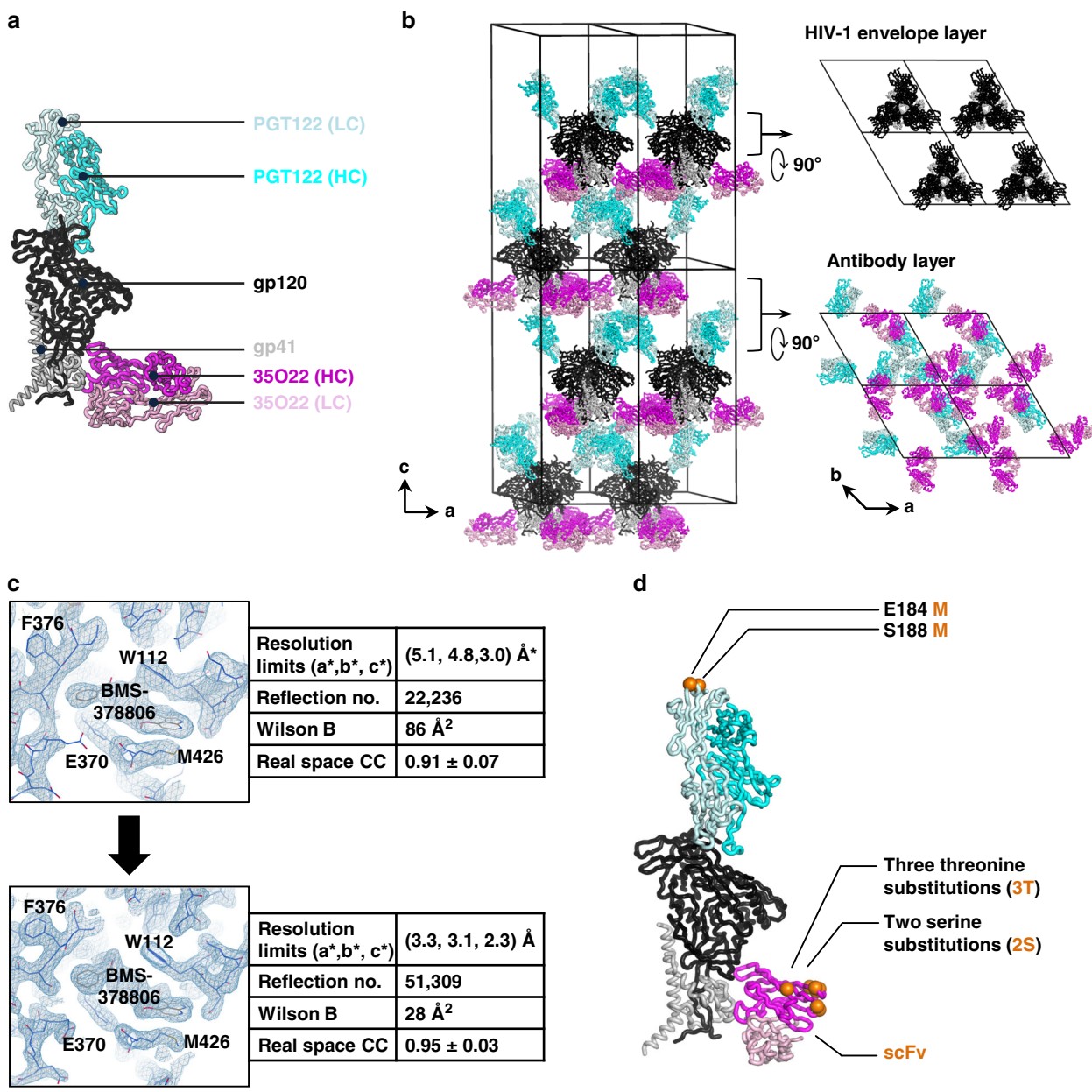

**Fig. 2** Molecular assembly of an HIV-1 Env lattice with improved diffraction. **a** The asymmetric unit of the P6₃ lattice is shown with the PGT122 Fab colored in cyan (HC; heavy chain) and light cyan (LC; light chain); 35O22 colored in magenta (HC; heavy chain) and light magenta (LC; light chain). gp120 and gp41 of BG505 SOSIP.664 Env trimer are colored as black and gray respectively. **b** Crystal packing within the P6₃ lattice, highlighting the lack of crystal contacts between Env trimers (top right). **c** Maps and diffraction statistics before and after lattice engineering. 2Fo-Fc electron density maps of the BMS-378806 binding pocket, shown as blue mesh, are contoured at $2\sigma$. (*The original data were truncated with different criteria. For fair comparison, the structure factor was downloaded from the Protein DataBank and the same criteria applied for truncation.) **d** Summary of the best changes identified in the antibody fragments to improve the diffraction

BG505 trimer through the terminal hydroxyl group, with clear electron density defining the conformation of the **BMS-818251** tail functional group (Fig. 4a; right panel). In addition, residue D113 of Env forms hydrogen bond with the amide nitrogen on the tail of **BMS-818251**. The lack of electron density surrounding the tail functional group of the highly similar **BMS-814508** (Fig. 4b; right panel) suggested that this functional group can adopt multiple conformations and lacks stabilizing interactions with Env.

The difference in the interaction between Env trimer and the tails of **BMS-818251** or **BMS-814508** likely explains the difference in their neutralization potency. Interestingly, although the chemistry of **BMS-818251** and **BMS-814508** was identical

beyond the tail functional group, we did observe additional interactions between Env residues F382 and E370 with **BMS-818251**, but not with **BMS-814508** (Fig. 4a, b, middle panels). Finally, we observed the piperidine ring of **BMS-818251** to adopt a twisted-boat conformation (Fig. 4a, b, left panels), with slightly higher energy than the chair conformation adopted by **BMS-814508**, suggesting a difference in tail interactions to affect the conformation of the entire compound.

**BMS-818251 accommodates residue variations with high potency.** BMS-818251 showed superior neutralization potency against the 30-strain panel compared with **BMS-814508**, ranging

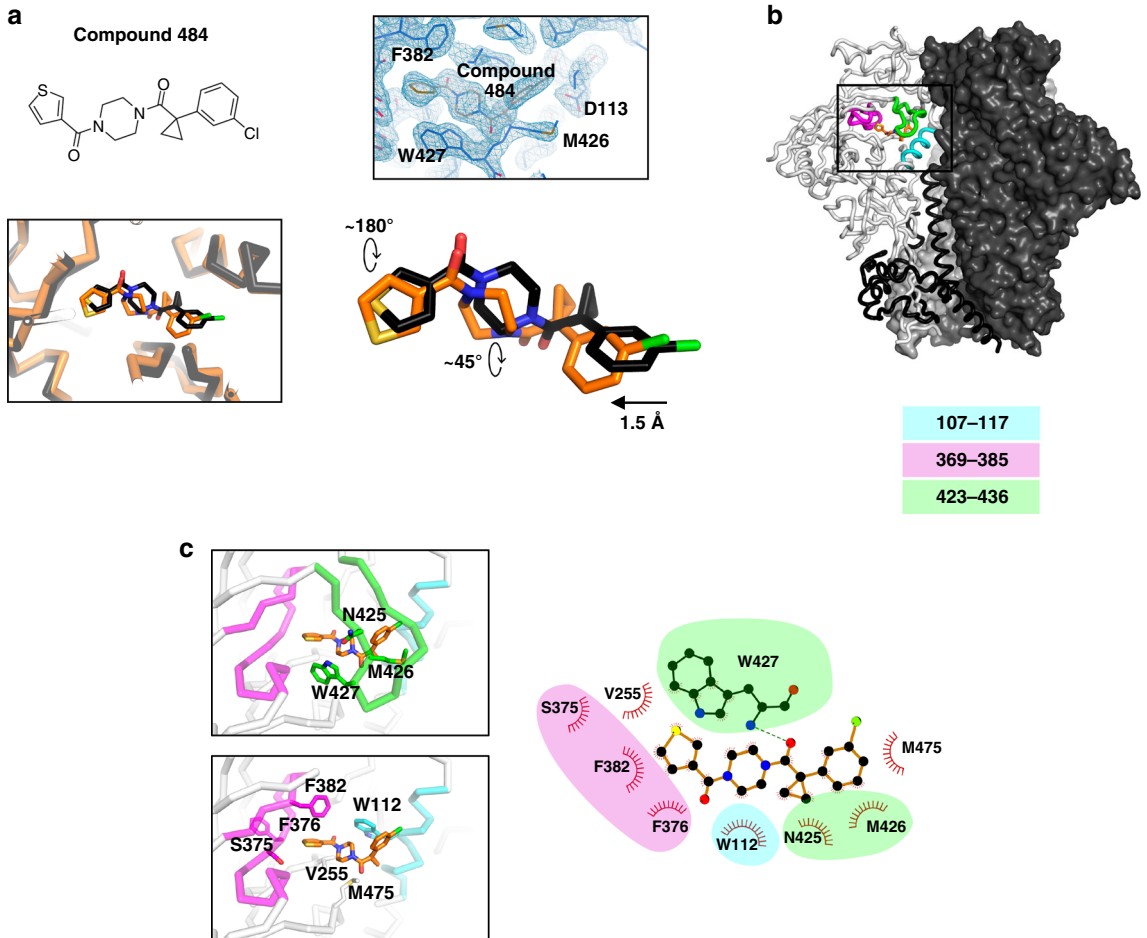

**Fig. 3** Chemical definition of compound 484 interaction with HIV-1 Env. **a** Comparison of cocrystal structure of **484** with docking simulation. The two structures are superimposed by the gp120 (**484** not used for superimposition). Simulated **484** complex structure is colored in black and co-crystal structure colored in orange (lower left). 2Fo-Fc electron density showing the **484** binding pocket is contoured at 2σ (top right). **b** Location of the entry inhibitor binding pocket in the context of HIV-1 Env trimer. The black square indicates the location of the inhibitor binding pocket. Three main protein segments interacting with the inhibitors are highlighted in cyan, magenta and green for residues 107–117, 369–385 and 423–436 respectively. **c** Detailed view of the **484** binding site. Upper panel on the left shows the residues of the β20−β21 hairpin (green fragment) interacting with **484**. Lower panel on the left shows the all other residues interacting with **484** (excluding residues on the β20−β21 hairpin, which is removed for clarity). Right, the **484**-Env interaction is shown as a 2D plot, with residues belonging to different major segments highlighted in different shades. Hydrophobic interactions are shown as sunray symbols while hydrophilic interactions are shown as dotted lines

from ~2.7- to 90-fold higher (Supplementary Fig. 7a; excluding three strains, DJ263.8, T253-11, and CAP256.206.C9, that were below the detection limit of assay for both **BMS-818251** and **BMS-814508**). Although the different tails of **BMS-818251** and **BMS-814508** induced subtle differences throughout their Env-bound conformation, we nonetheless focused on Env residues 117, 429, and 432 to provide an explanation for the wide range of improved potency of **BMS-818251**, because of the direct inter-action between these residues and the **BMS-818251** tail in the BG505 trimer cocrystal structure (Fig. 4a).

Residue 117 was Lys in all 30 tested HIV-1 strains (except for strain CNE56, where it is a Gln), with multiple residue types at residues 429 and 432; we therefore categorized the fold improvement based on the residue identity in residues 429 and 432 (Supplementary Fig. 7b). With the most prevalent combina-tion of R429 and Q432 (9 out of 30 strains), the improvement in neutralization potency for **BMS-818251** ranged from ~2.7 to ~50 fold. With G429 and a positively charged residue 432 (K or R; 5 out of 30 strains), the improvement ranged from ~3.3 to ~53.3-fold, similar to the R429 and Q432 combination. With negatively charged E429 and a positively charged residue 432 (K or R; 8 out

of 30 strains), the fold improvements were similar compared to the combination of R429 and Q432 (ranging from ~3.3 to 90-fold). Finally, when the residues at 429 and 432 were both positively charged (R429/K432 for REJO.67 and K429/K432 for BL01.DK), the improvement were generally minimal (6.1-fold and 5.3-fold, respectively). Thus a wide range of neutralization sensitivity was observed even with identical or homologous residues at positions 429 and 432. However, we noted that the neutralization potency of **BMS-818251** improved for most of the viruses tested. Overall, the virus neutralization data supported the general benefit of the tail functional group in **BMS-818251** compared to its counterpart in **BMS-814508**, and revealed that improved potency could generally be achieved despite variations at positions 429 and 432, which directly interacted with the **BMS-818251** tail functional groups.

**Structural basis of clade-dependent neutralization.** We sought to examine whether the new lattice could provide experimental feedback on the recognition of Env trimers from different HIV-1 clades. Clade-dependent sensitivity has been observed for the entry inhibitors in the **BMS-626529** family[18]. Among the three

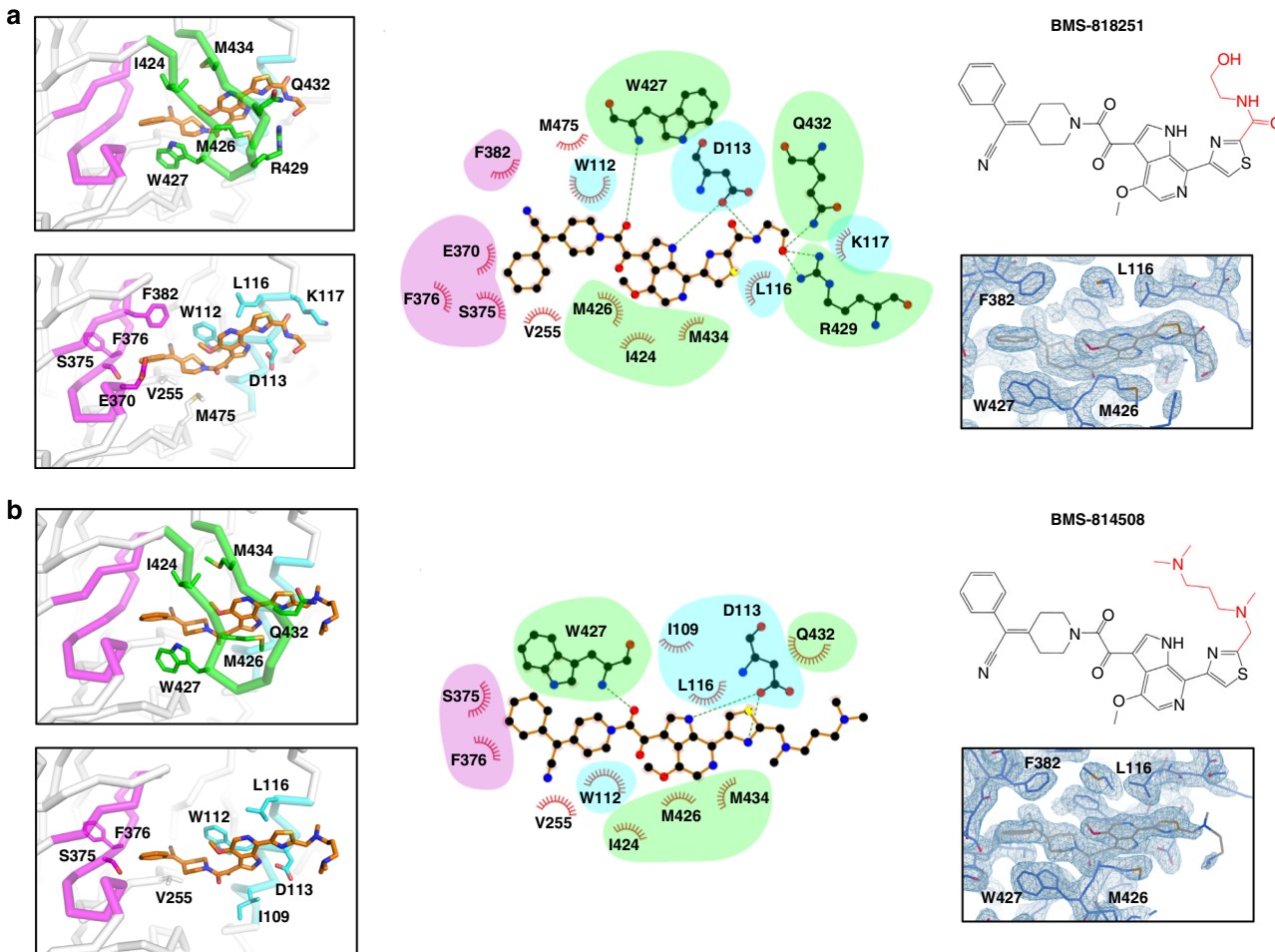

**Fig. 4** Crystal structures derived from improved lattice reveal molecular features associated with enhanced neutralization potency. **a** The complex structure of **BMS-818251** with BG505 SOSIP.664 Env trimer is shown in the top panel on the left with interacting residues on the β20−β21 hairpin (green segment) shown as sticks and labeled; interacting residues other than those on the β20−β21 hairpin are shown as sticks and labeled in the lower left panel. **BMS-818251** is shown as orange sticks. Middle, **BMS-818251** interaction with BG505 SOSIP.664 Env trimer is shown as a 2D plot, with residues belonging to the three major fragments colored accordingly. Top right, the chemical formula of **BMS-818251**; bottom right, 2Fo-Fc electron density map surrounding the inhibitor binding pocket is shown as blue mesh, contoured at 2σ. **b** Crystal structure (left), 2D plot (middle) and electron density map (bottom right) of **BMS-814508** (top right) shows the interaction between the inhibitor and BG505 SOSIP.664 Env trimers

major clades that had been well studied with this class of inhibitors, clade B strains were the most sensitive, followed by clade C strains with moderate sensitivity and clade A strains with least sensitivity. To investigate the structural attributes responsible for this clade dependency, we first determined the compound-free Env-trimer structure of the B41 strain, a clade B virus with only 78% sequence identity to the clade A BG505. Residues that differed in sequence between B41 and BG505 were observed to account for 28% of the Env-trimer surface (Fig. 5a). Surface differences were also present in the epitopes of the crystallization chaperones 35O22_3T2S and 3H109L_MM, but these modified antibodies were able to accommodate these changes, and crystals of the B41 SOSIP.664 Env trimer had unit cell dimensions similar to those of the BG505 SOSIP.664 Env trimer crystals (Fig. 5b).

We next sought to determine cocrystal structures of Envs from HIV-1 strains B41 and BG505 with **BMS-386150**, a compound similar to the prototypic **BMS-378806**, though with a bromine modification to aid structure determination (Fig. 1a). Although there were differences in the three major binding fragments (Fig. 5c, colored in cyan, magenta and green respectively, as defined in Fig. 3b) interacting with **BMS-386150**, the interacting residues were mostly conserved between B41 and BG505 Envs.

Furthermore, the **BMS-386150**-interacting residues specific for B41 or BG505 (highlighted in black rectangles in Fig. 5c) were conserved between these two strains (Supplementary Fig. 8). This observation suggested that the clade dependency of entry inhibitors in the temsavir family was not directly caused by sequence variation in the binding pocket, but might rather be a consequence of various degrees of conformational pliability at the inhibitor binding site among the different clades. Indeed, a modest but significant shift of the β20−β21 hairpin loop was observed in the **BMS-386150**-complexed structures of B41 and BG505 Envs (Fig. 5d, upper panel), supporting the idea that conformational changes around the binding pocket, instead of the sequence identity of residues of the binding pocket, dictate the clade-dependent neutralization potency of this class of entry inhibitors. This conclusion was further strengthened by the fact that B41 and BG505 showed conformational differences around the drug-binding pocket in their compound-free structures (Fig. 5d, lower panel).

## Discussion

Crystal engineering has been extensively explored for small molecules crystals[26,27], such as metal-organic frameworks[28,29],

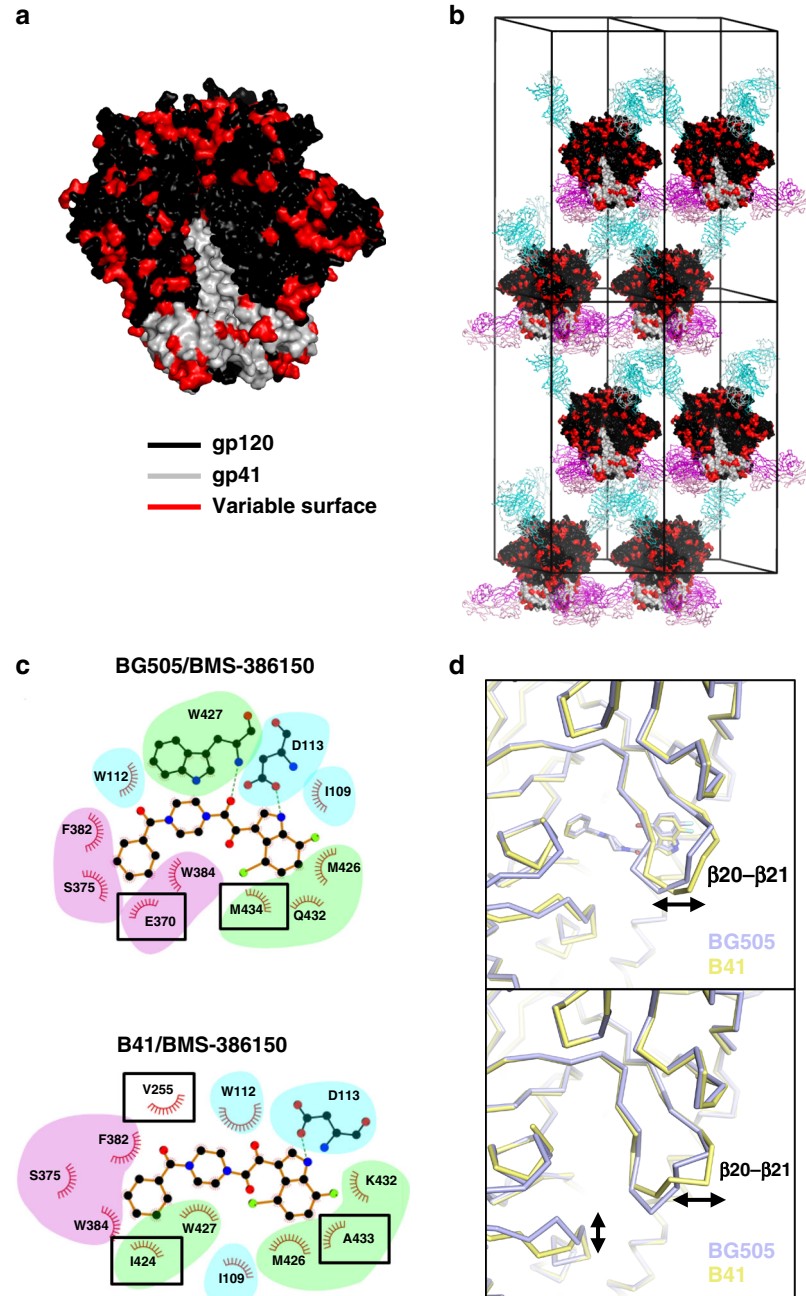

**Fig. 5** Generalizable crystallization platform for diverse HIV-1 envelope and drug complexes. **a** B41 SOSIP.664 Env trimeric structure in surface representation shows gp120 in black and gp41 in gray; the residues that differ between the B41 SOSIP.664 and BG505 SOSIP.664 Env trimers are colored as red. **b** B41 SOSIP.664 Env trimeric structure is shown in the context of P6$_3$ lattice. **c** Structures of **BMS-386150** in complex with the BG505 (top) or B41 (bottom) SOSIP.664 Env trimers. Residues interacting with **BMS-386150** that are different between B41 and BG505 are highlighted in black box. **d** The backbone conformational differences in the structures of **BMS-386150** complexed with BG505 (purple) or B41 (yellow) are shown as ribbon diagram (top), the same region in the apo-form BG505 and B41 are also shown (bottom)

which has led to crystals with extraordinary properties, including those with lattice cages, capable of holding diverse small molecules[30,31]. Crystals of biological macromolecules may be more difficult to manipulate due to their greater complexity; however, recent advances have allowed crystal engineering of biological crystals of DNA, protein and peptides[32–34]. We report here the engineering of a protein crystal composed of a highly glycosylated trimer and two crystallization chaperones, which were adapted antibody fragments. We focused on modifying the

crystallization chaperones, to remove chaperone flexibility, while avoiding alterations to the target protein. Of note, the antibody fragments were resilient to alteration, and for one of the crystallization chaperones (35O22 Fab), we removed half of its weight, while increasing its capacity to facilitate crystallization. The resulting engineered lattice led to higher resolution and yielded twice as many unique measured reflections as the original crystal, while maintaining similar unit cell parameters; this allowed for improved electron density and more accurate

modeling of small-molecule inhibitors and Env trimers. Overall, the lattice engineering we report is conceptually similar to a retro-engineering strategy reported recently for an organic molecular crystal[35], but has not to our knowledge been used for macromolecular lattice improvement.

Although antibody fragments and other crystallization chaperones are popular tools for crystallizing difficult protein targets[36,37], the retrospective engineering of crystallization chaperones has not been extensively explored. More and more protein targets are being crystallized in the presence of crystallization chaperones[38,39], as many of these crystals diffract poorly due to suboptimal chaperone properties[40,41], the engineering strategy reported here may represent a straightforward option for improving their diffraction.

The antiviral potency of the fostemsavir inhibitor is known to display HIV-1 clade dependence on neutralization[18], but the molecular basis for this dependence has not been identified. Interestingly, we observed conformational differences in the interaction of **BMS-386150** with Env trimers from clade A BG505 and clade B B41 HIV-1 strains, despite conservation of all Env residues that contacted **BMS-386150**. Differences between these interactions related to the positioning and orientation of contact residues, which differed despite being identical in sequence. Specifically, we observed the BG505 and B41 Env trimers to adopt slightly different conformations in their respective binding pockets around the $\beta 20-\beta 21$ hairpin loop region. This difference in binding pocket conformation led to different conserved residues interacting with the inhibitor, manifested as clade dependency in neutralization assays. Our results thus provide insight into the complexity of clade dependence and reveal the importance of obtaining experimental information on small molecules in complex with target Env trimers from diverse clades. The crystal structures of **BMS-386150** also provided a basis for its reduced potency, as the bromine substitution, despite being beneficial for structure determination, abrogated the hydrophobic interactions between I424, M426 and M434 and the methoxy functional group that was replaced.

Despite the general importance of clade dependency, we note that the tail functional group that endowed high potency to **BMS-818251** could accommodate variations on interacting residues at positions 429 and 432 of the HIV-1 Env. The effects of variation in these residues on the enhanced potency of **BMS-818251** did not show clade dependency (Supplementary Fig. 7b). The flexibility of the tail functional group of **BMS-818251**, which can adopt different conformations through torsional angle freedom, may allow it to accommodate the different conformations of the $\beta 20-\beta 21$ structural element in different clades. Furthermore, we note that the **BMS-818251** tail contains two functional groups that can be either hydrogen-bond donors or acceptors, potentially increasing its versatility to interact with HIV-1 Env, while the **BMS-814508** tail contains two nitrogens that can only be hydrogen-bond acceptors. Thus, multiple ways may exist for small molecules to succumb to or to overcome clade dependency.

ITC binding data showed that **BMS-818251** has a ~10-fold higher affinity to BG505 DS-SOSIP.664 trimer compared to **BMS-626529** binding, which is in good agreement with the overall >10-fold improved neutralization potency of **BMS-818251**. However, we note that the neutralization potency is higher than the binding affinity measured by ITC (for example, the BG505 virus was neutralized at 0.0003 μM by **BMS-818251**, while the ITC-determined binding affinity between **BMS-818251** and BG505 DS-SOSIP.664 Env trimer was 0.047 μM). This discrepancy is likely because BG505 DS-SOSIP.664 Env trimer is not a perfect mimic of the native BG505 Env trimer on the virion surface. Nonetheless, the fold improvement in affinity correlated well with the fold improvement in neutralization. Thus, BG505

DS-SOSIP.664 Env trimer, despite being an imperfect mimic of native Env, can be a valuable surrogate for the evaluation of improved next-generation entry inhibitors.

It is noteworthy that **BMS-818251** exhibited higher antiviral potency than **BMS-814508**, a related compound that differs only in the tail functional group, for all 30 viruses (excluding 3 viruses that were below the detection limit of assay) tested in this study (Supplementary Fig. 7). AE strains generally are less sensitive to temsavir but were sensitive to **BMS-818251**, suggesting that **BMS-818251** could potentially be pan-reactive to HIV-1 Env. Indeed, we observed a general improvement for **BMS-818251** across all major HIV-1 clades, likely enabled by its novel tail functional group. Further optimization of this tail functional group may lead to inhibitors with greater resistance to viral escape. We note that the prodrug version of temsavir is one of the few gp120-directed therapeutics being assessed clinically, with gp120-reactive antibodies like antibody VRC01 in clinical assessment showing lower breadth and potency than temsavir (Supplementary Fig. 9). Overall, the structure of the **BMS-818251**-Env complex enabled by chaperone-based crystal engineering reveals molecular features explaining the superior neutralization potency of this compound and assisting in the rational design of next-generation inhibitors targeting the gp120 component of the HIV-1 Env.

## Methods

**Synthesis of small-molecule entry inhibitors**. The BMS entry inhibitors were synthesized based on Supplementary Figure 10. The oxalamides were prepared via acylation of indole with oxalyl chloride followed by coupling of the acid chloride with 1-benzoylpiperazine (ref. [42]) or azaindole with methyl chlorooxacetate in the presence of AlCl₃, followed by hydrolysis of the methyl ester and coupling with 1-benzoylpiperazine (ref. [43]) or 2-phenyl-2-(piperidin-4-ylidene)acetonitrile. The substitutions on C-7 of 6-azaindole were installed initially via either copper(0)-mediated radical reaction or palladium(0)-mediated Stille coupling, followed by structural modifications. Inhibitors were characterized by 1H-NMR, 13C-NMR and high-resolution mass spectrometry after synthesis; the representative spectra of **BMS-818251**, the most potent compound reported in this manuscript are shown in Supplementary Fig. 11. Compound **484** were prepared following published protocol[20] and described here. N,N-diisopropyl ethylamine (0.6 mmol), 4-dimethylaminopyridine (0.03 mmol) and N-(3-dimethylaminopropyl)-N′-ethyl-carbodiimide hydrochloride (0.45 mmol) were added to a solution of carboxylic acid (0.3 mmol) and amine (0.3 mmol) in anhydrous dichloromethane (2 ml). The reaction was stirred at room temperature for 5 h and subsequently concentrated in vacuo. The resulting residue was purified by flash chromatography and the purity of Compound **484** was greater than 95% according to NMR analysis.

**HIV-1 neutralization assays**. Neutralization was measured using single-round-of-infection HIV-1 Env-pseudoviruses and TZM-bl target cells (ARP Cat No. #8129)[44,45]. Compounds were prepared at 400 μM with 4% DMSO (dimethyl sulfoxide) in PBS (phosphate-buffered saline) and diluted 20 times as the starting concentration for serial dilution in the neutralization assays. Pseudoviruses of ten strains from HIV-1 clades A, B, and C were incubated with serial dilutions of drug compounds, then added to TZM-bl target cells which have a luciferase reporter gene. After incubation at 37 ℃ for 2 days, infection of target cells was measured by luciferase activity. Neutralization curves were fit by nonlinear regression using a five-parameter hill slope equation[45].

**Proteins expression and purification**. BG505 SOSIP.664 is a construct with modifications that enabled the production of fully cleaved, prefusion-closed, soluble HIV-1 Env trimer[21]. Proteins were purified following published protocols[24] with modifications described below. For 1 L of Gnti⁻/⁻ cell culture (ATCC Cat No. CRL-3022), 600ug of BG505 SOSIP.664 (with N332 glycan) expression plasmid and 150 μg expression plasmid for Furin were diluted in 50 ml Opti-MEM medium (Life Technology). Three milliliters of Turbo293 transfection reagent (Speed Bio-Systems) was diluted into 50 ml Opti-MEM medium and incubated for 5 min. Diluted transfection reagent was then added into diluted DNA and incubated for 15 min. The transfection reagent and DNA complex were added into 800 ml of Gnti⁻/⁻ cells at 2 million/ml. On the second day of transfection, added 80 ml HyClone SFM4HEK293 medium to each flask of 1 l cell cultures. Return flasks to shaker incubator at 120 rpm, 37 ℃, 9% CO₂ for additional 6 days. Supernatant was affinity-purified over 2G12 column, eluted by 3 M MgCl₂, followed by size exclusion chromatography in 5 mM Hepes and 150 mM NaCl. Variants of 35O22 and 3H109L antibody fragments were produced in Gnti⁻/⁻ and Expi cells respectively following published protocol[24]. 3H109L variants were produced as Fab, with heavy chain truncated at the end of constant domain 1 and a purification tag containing

thrombin site, Strep-tag and 6xHis-tag was appended. After tandem purification by Strep column and Ni-NTA column, the tag was removed by thrombin digestion at 4 °C overnight, followed by size exclusion chromatography. 35O22 scFv has a single 6xHis-tag for purification and the tag is retained after purification. Two artificial glycans introduced to 35O22 were preserved by not treating with endoglycosidase.

**Crystallization and data collection**. Purified trimers were mixed with 3H109L Fab variant (with two methionine modifications) and 35O22_scFv variant (with 3T, 2S and 2 glycan modifications) in a 1:3.2:3.2 molar ratio (gp140 protomer: Fab:scFv) and incubated overnight at room temperature before being purified by SEC and concentrated to 10–15 mg/ml for crystallization. Glycans on the Env and antibody fragments were preserved without the treatment of glycosidase. Crystallization conditions were obtained by sparse chemical screening and optimized manually after initial hits were identified. Diffraction data were collected at APS ID22 and processed using HKL2000[46]. The overall resolution was determined as the highest resolution for which the completeness was greater than 50% and the I/σI was >2.0. Data were then truncated by UCLA anisotropy server (https://services.mbi.ucla.edu/anisoscale/).

**Structure solution and refinement**. The structures of entry inhibitors in complex with BG505 or B41 SOSIP.664 in the presence of crystallization chaperone variants were solved by molecular replacement using Phaser in CCP4[47] using the nondrug-complex structure as search model, PDB ID 4TVP[24]. Small molecules were manually fitted using COOT[48] and refinement of the structures was done in Phenix[49] using optimization of X-ray/stereochemistry and X-ray/ADP weight. CIF files for the small molecules were obtained using ELBOW in Phenix. The asymmetric unit of each crystal structures contains one copy of gp140 protomer, one copy of 35O22 scFv and one copy of 3H209L Fab, with one small-molecule inhibitor occupying the gp120 binding pocket (except the B41 apo structure). Simulated annealing Fo-Fc difference omit map (with small-molecule inhibitors omitted during map calculation) confirmed the modeling of small-molecule inhibitors in the binding pocket (Supplementary Fig. 12). Stereo images of the final 2Fo-Fc map, overlaid on the final model, were shown in Supplementary Fig. 13. The refinement statistics are summarized in Supplementary Table 4. Structural figures were made with Pymol[50].

**Binding affinity measurement**. The binding of **BMS-626529** and **BMS-818251** to BG505 DS-SOSIP was studied by ITC using a VP-ITC from MicroCal/Malvern Instruments Ltd. (Northampton, MA). Prior to the experiment, BG505 DS-SOSIP was dialyzed against PBS, pH 7.5. The inhibitors were first dissolved in 100% DMSO at concentrations of 10 mM, which were then diluted into PBS with additional DMSO to their experimental concentrations of 40 μM in the presence of 4% DMSO. Titrations were performed at 37 °C by injecting 10 μl aliquots of the inhibitor solution into the calorimetric cell (volume ~1.4 ml) containing BG505 DS-SOSIP prepared at ~2.2 μM (expressed per protomer) in buffer with 4% DMSO.

## Data availability

Crystal structure data that support the findings of this study have been deposited in the Protein Data Bank under the accession code 6MTJ, 6MTN, 6MU6, 6MU7, 6MU8, 6MUF and 6MUG. All other data generated or analyzed during this study are included in this published article (and the supplementary files) or are available from the corresponding author on reasonable request.

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

## Acknowledgements

We thank D. Burton and M. Fineberg for the sequences of NAC-identified antibodies, including the 3H109L member of the PGT121 lineage used in crystallization, M. Connors for the sequence of antibody 35O22 used in crystallization, J. Stuckey for assistance with figures, and members of Structural Biology and Structural Bioinformatics Core Sections, Vaccine Research Center for discussions and comments on the manuscript and the WCMC/AMC/TSRI HIVRAD team for their contributions to the design and validation of near-native mimicry for soluble BG505 SOSIP.664 trimers. Funding was provided by the Intramural Research Program of the Vaccine Research Center, National Institute of Allergy and Infectious Diseases, the Intramural AIDS Targeted Antiretroviral Program, the National Institute of General Medical Sciences, National Institutes of Health. Use of sector 22 (Southeast Region Collaborative Access team) at the Advanced Photon Source was supported by the US Department of Energy, Basic Energy Sciences, Office of Science, under contract no. W-31-109-Eng-38.

## Author contributions

Y.-T.L. determined the crystal structures with assistance from D.R.L. and P.D.K. S.O., M. K.L., B.L., K.M., R.T.B., N.A.D.-R. and J.R.M. performed virus neutralization assays. A.S. performed and analyzed ITC experiments. C.S.C., A.D., D.P., Y.Y. and B.Z. expressed and purified proteins. T.W., A.H. and J.S. synthesized small-molecule compounds. G.-Y. C. analyzed virus resistance data. Y.-T.L. and P.D.K. analyzed the data and wrote the paper.

## Additional information

**Competing interests:** D.R.L. and T.W. own stock in Bristol-Myers Squibb. All other authors declare no competing interests.

