## [Peer Review File · Nature Communications]

Reviewers' Comments:

Reviewer #1:

Remarks to the Author:

The manuscript entitled "An engineered crystallization lattice reveals molecular features allowing for potent small-molecule inhibition of HIV-1 entry" by Lai et al. reports multiple crystal structures with and without small molecule entry inhibitors binding to a gp120 binding pocket that show improvements over previous in class candidates. This study extends previous results on the same class of entry inhibitor small molecules and report improved potency over the previous published results from the same groups. The paper contains interesting and important data that would be useful for structure-based drug design. However, the manuscript requires substantial revision to more adequately assess the inferences drawn from the experiments and analyses.

Main Comments

General: Was the 20-fold improvement in potency also accompanied by an observed improvement in binding for the two identified compounds? There are no binding data shown for the small molecules being discussed and it would be useful to perform binding experiments against a few representative isolates from each of the different clades for reference, preferably from those on the virus panel to assess trends between potency and binding and if clade-specific behavior is apparent. The engineering of lattice contacts is of some importance for this HIV-1 envelope protein as it has been quite difficult to generate high resolution data on Env complexes with antibodies or small molecules. So this in itself is a great achievement. However, in my opinion, the language is a little loose in the title and elsewhere about the lattice engineering where it is the structural information derived from crystals with the engineered lattice that is providing new insights not the lattice engineering per se.

Line 6-7. The title is misleading as the lattice does not allow for potent small molecule inhibition of HIV-1 entry nor do molecular features per se.

Line 37. "define accurately". Do you mean at higher resolution??

Lines 40-41. "and provided a structural explanation for the improved neutralization of BMS-818251" The engineered lattice did not strictly provide this explanation.

Line 43. "Crystal lattice engineering thus provide insights" Sorry to keep being picky here but the lattice engineering did not provide these insights, the structures obtained from the crystals derived the crystal lattice engineering provided these.

Line 66. Please replace "then altered" with "then be altered".

Lines 90-94 and 116-118 mention that BMS-814508 has a 4-fold improvement in potency compared to BMS-626529. Is this true after testing against this minimal virus panel or is it the same as BMS-626529? Lines 90-94 and Line 116-118 seem to have conflicting statements. Is the mean inhibition much lower on a 208-virus panel?

Line 91. Please replace "improve entry" with "improved entry".

Line 91-92. So is the HIV-1 NL4-3 strain an easily neutralized tier1 virus?

Line 99. This is a major experiment on which the punchline of the paper is based. However, it is unclear why only 30 strains were tested instead of the 208-virus panel to infer potency improvements over that reported by Pancera et al. in Nat Chem Biol. To clearly categorize these new small molecules as better leads in the category of gp120 inhibitors, the comparisons really should be head to head,

and it is recommended that the compounds be screened against the identical 208-virus panel reported in the previous study, to remove all doubts of strain-specific improvements, unless a some further rationale is given here of why only the 30-strain panel was used.

Line 103. What was the basis of the engineered modifications in the compounds that were designed to aid crystal structure determination?

Line 107. Please replace "componenet" with "component".

Line 109. Neutralization of the 30-strain panel generally recapitulated prior observations. What was the rationale of including these 30 HIV-1 strains on this panel? Were these the most difficult to neutralize isolates in the study with BMS-626529 that are now being neutralized by the small molecules in this study?

Lines 114-115. The modification mentioned on Line 103 to aid structure determination reduces the neutralization potency of BMS-386150. The authors should elaborate on the structural basis for this observation from the Env complex structure at 2.93 Å resolution.

Line 120. Please delete "higher".

Lines 138-140. Is this replicable across isolates from most clades autologous Envs to demonstrate generalizability rather than confined to Envs from strains such as BG505 and B41 that are already known to be favorable for crystallization.

Lines 141-142. "context, mainly due to the lack of structural support of antibody constant region." I am not sure I fully understand this as the elbow angle is quite flexible and the constant regions are often not well resolved in crystal structures. So where does the 'lack of structural support' come from?

Line 145. Please replace "28.3 Å²" with "28 Å²".

Line 146. Please replace "85.6 Å²" with "86 Å²".

Lines 146-148. Can this also be a stabilization effect of these new small-molecules studied in this work compared to the 484 and BMS-626529 molecules? The claim to generalizability could be supported with a similar resolution structure without the small molecules bound to show the direct effect of lattice engineering with altered chaperones, which is not presently available.

Line 147. Please replace "71.2 Å²" with "71 Å²".

Line 147. Please replace "201.5 Å²" with "201 Å²".

Line 158. "the unique diffraction number" Is this the number of unique reflections measured?

Line 160. Please replace "the information" with "the effective information".

Line 160-161. "constraints obtained from the new crystal more than doubled". I am not sure what is meant by the constraints doubling?

Line 151. This is really interesting but where are these two glycosylation sites on the 35O22 variants?

Line 152. This figure does not mention where the two artificial PNG sites were introduced to improve the yield 10x. Figure S2b simply repeats that 2 PNGs have been introduced.

Line 156. Since most known crystal structures of HIV-1 suffer from anisotropy, showing a small section of the structure does not represent the full-picture. The authors should mention if the disordered regions, for e.g., the HR1N etc. is more ordered in the improved lattice. Also, omit maps should be provided for an unbiased evaluation of the improvement due to lattice engineering. The figures do not mention what the illustrated maps are- 2Fo-Fc or omit maps? These need to be clearly labeled in the fig legends. Also, which region is being shown in Fig 2c? There are no labels and no reference to comprehend the orientation.

Lines 163-168. Are these two regions being contacted by the small molecules? Or this improved resolution can unambiguously be mapped solely to the improvement in lattice contacts by engineering the antibodies?

Line 176. It is unclear how the Env complex with 484 adds to the overall scientific impact of the manuscript.

Line 180. Where is the crystallographic table for this co-crystal structure? This should be provided.

Line 184. "clear electron density". It is not clear what electron density map is shown in Fig. 3a. Please show the difference Fo-Fc map after molecular replacement or omit map for molecule 484. Also, the protein should be depicted in cartoon/ribbon in a single color and the small molecule shown in a separate figure panel for clear comparison.

Line 185. It would be nice to have zoomed figure showing superimposition of docked and crystal structure conformation of compound 484. Also, show zoomed omit map for compound 484.

Line 190 and throughout. IC50 should be IC50.

Lines 200-201. "whether the engineered lattice could provide a structure-function explanation" Sorry, but again the engineered lattice doesn't give that information but the diffraction data acquired from the crystals with the engineered lattice.

Line 213. Is this BMS-814251 or BMS-818251?

Line 214. The authors infer that the flexibility and therefore lack of contacts with the tail region of BMS-814508 relates to its reduced neutralization potency compared to BMS-818251. What the authors have not commented upon in relating structure-to-function is which contacts in BMS-818251 causes the 25-fold increase in potency compared to BMS-814508 and which contacts are responsible for the cytotoxicity observed at high concentrations of BMS-814508?

Line 218. Please replace "twist" with "twisted".

Line 218. Is this piperidine or piperazine ring? Does BMS-818251 have a piperazine or piperadine (Fig. S1)?

Line 218-219. The cause of these geometric constraints in conformation need more explanation? Perhaps a search for this conformation in the Cambridge crystallographic database would be appropriate to check the conformation?

Line 220. Which are the interactions in the small molecule core or tail that cause the huge difference in potency between these two molecules?

Line 222-223. Overall, it is hard to make any conclusion out of these Env residues in relation to improvement in potency.

Line 224. This should also if possible be tested against the identical 208-virus panel to support this inference.

Line 229. Why not L116? From figure 4a, it seems L116 is also interacting with the tail of BMS-818251.

Line 232. What is the residue conservation score at positions 117, 429 and 432 of the Env across all recorded unique isolates?

Line 253. It might have been mentioned elsewhere, but it would be useful to put results in to context: Were only 3 clades - A, B and C tested? If yes, then what were the particular reasons to be limited to these 3 clades?

Line 259. Is this the value for only one out of the three gp140 component of the trimer or is this value for the entire trimer?

Line 260. "Surface differences were also present in the epitopes of the crystallization chaperones". Please describe the differences. Was it only sequenced-based? Or are there corresponding noticeable local conformational differences?

Line 263. Fig 5b. Is this with or without the small molecules?

Line 267. "differences in the three major binding fragments ...". Please describe these in more detail in the manuscript.

Line 272. "consequence of various degrees of conformational pliability..." Does the comparison with the previously published B41 (Barnes et al., Nat Commun, 2018) and BG505 structures, without small molecules bound, justify this hypothesis of local conformational change in this region? Or is this only a consequence of small-molecule binding?

Line 276. Please replace "dictat" with "dictate".

Line 292. "while increasing its capacity to facilitate crystallization" Did this result in more crystals from different conditions or solely from an increase in resolution?

Lines 293-294. "engineered lattice yielded twice as many unique reflections as the original crystal, while maintaining similar unit cell parameters". Sorry but this seems to be due to the increase in resolution if the lattice parameters are effectively identical.

Lines 297-298. "but has not to our knowledge been used for macromolecular lattice improvement." Have lattice contacts not been engineered previously in protein crystals? I would have thought they have. So how does this approach differ from those?

Lines 311-312. "trimers adopt slightly different conformations" This contrasts with Line 273 where the authors infer substantial difference in conformation. Which is correct? This should be rephrased to

something similar to "the binding pocket for this set of small molecule entry inhibitors around β 20- β 21 hairpin loop region".

Lines 327-328. What is the experimental basis of this inference?

Line 340. "superior neutralization potency" This should only be claimed/validated if the small molecules are screened over an identical 208-virus panel tested for BMS-626529.

Lines 361 and 382. "37°" "4°" Please correct to the correct unit of temperature throughout the text.

Line 386. It is unclear if the 35O22scFv was deglycosylated after expression and affinity purification and prior to crystallization screens to remove the expression-enhancing artificially-introduced PNGs. The authors should clarify in the methods where these PNGs were introduced and how the scFv was treated post-expression.

Line 387. Which variants were used for complex formation out of the four in Supplementary Figure 2b? "1:3:3 molar ratio (gp140 protomer:Fab:scFv) and incubated overnight at room temperature" Was there no slight molar excess of any of the antibodies?

Line 389. What about carbohydrates of Env? Were complexes treated with glycosidases or not? Which 35O22 variants were used, with or without glycosylated variants?

Line 393. Please replace "Data is" with "Data were then".

Line 402. The crystallography table does not mention clearly the following: What are the components of the complex and how many copies of each component is present in the complex.

Line 430. "Journal of medicinal chemistry (2017)" Incomplete reference.

Line 458. "Nat Chem Biol (2017)". Incomplete reference.

Lines 482-483. "Proceedings of the National Academy of Sciences of the United States of America" Please abbreviate journal reference.

Figure 2. A side-by-side comparison of the complexes with Fabs and scFvs would be helpful.

Figure 2C. Please replace "85.6 Å²" with "86 Å²". Integers are sufficient for Wilson B and other B factors.

Figure 2, Legend, line 4. Please replace "P63" with "P6₃"

Figure 3, Legend, line 1. "Accurate". I do not think accurate is the right word.

Figure 4, Legend, line 1. "Improved lattice" The structure reveals these features not the improved lattice.

Figure 4, Legend, line 7-8. "electron density map" What type of map - 2Fo-Fc??

Figure 5, Legend, lines 6-8. "The backbone conformational differences in the structures of BMS-386150 complexed with BG505 (green) or B41 (cyan) are shown as ribbon diagram." What are the in

the top and bottom panels?? They are different but appears to be the same complexes and the same region?

Figure 3. Please depict the protein and small molecule differently (e.g. cartoon/ribbon and ball-and-stick) throughout the manuscript for clear identification.

Figure 5d. Could the sequence alignment of this region be included to also show differences or similarity?

Supplementary Text

Table S2. Are these complexes with Env and Abs? If yes, please mention it in header of table. Rpim values are missing for all structures. It would be nice to show omit map for each molecule used in complex.

Figure S3. The B41 complex with the engineered scFv does not retain the lattice shown in Barnes et al. Nat Commun, 2018. How does this technique relate to being generalizable? Also, this study lacks the comparison with the Barnes et al. structure with the Fab components at both epitopes with the B41 structures with scFvs reported in the present study.

Why these numbers of ratio of reflections to atoms cited may in some respects be technically correct, it is a little misleading when some atoms have been removed and the lattice remains the same. I am not sure how comfortable I am about the possible interpretations from these numbers made on such an analysis.

Figure S4. "5CEZ". What are the extra residues here in 5CEZ??

Figure S5. Please label residues causing the bulge and some others around it in the figure for orientation. Are there no differences to be observed in the B41 complexes with the Fabs versus scFvs? There should probably be some comparison with the Barnes et al. Nat Commun 2018 structure?

Page 14, Table S2

Rsym or Rmerge (%). "(143)" 143.0??

Please add number of unique reflections measured.

Resolution (Å) "2.34" "2.50" This is not the actual resolution of this structure- please use the accepted definition of effective resolution when a large number of reflections are missing from the data.

B-factors

Please replace "38.9" with "39".

Please replace "43.4" with "43".

Protein

Please replace "38.1" with "38".

Please replace "42.4" with "42".

Ligand/ion

Please replace "55.8" with "56".

Please replace "61.2" with "61".

Water

Please replace "23.26" with "23".

Please replace "31.1" with "31".

Need to say what the red values are. Some of the higher resolution numbers do not make sense compared to the lower resolution values.

Page 15, Table S2

Please add number of unique reflections measured.

I / σ I

Please replace "17.1 (2.0); 17.38 (1.2)" with "17.1 (2.0); 17.4 (1.2)".

Resolution (\AA)

This is not the actual resolution of this structure. Please use the accepted definition of effective resolution when a large number of reflections are missing from the data.

B-factors

Protein, Ligand/ion, Water

Please convert these to integers with no decimal points.

Reviewer #2:

Remarks to the Author:

This paper by Lai et al reveals a newly engineered crystallization lattice for HIV Env which allows for studying binding of small molecule HIV entry inhibitors at resolutions where the molecular details are more clearly visible. This engineered lattice framework is novel and the methods described here are of interest not only to the HIV community, but also applicable to other fields as well and should be of broad interest to the general scientific community. The work provided here is of high quality and very thorough. This work provides valuable insights into the molecular details of clade specific inhibition of HIV Env by small molecules.

Minor points:

1. The overall completeness is low for many of the crystal structures, particularly for B41+35O22_3T2S+3H109L_MM+BMS-386150 and BMS-386150. Could the authors comment as to why this may be?
2. There seems to be a typo in the number of protein atoms for B41+35O22_3T2S+3H109L_MM in the supplementary materials, please correct.
3. The labels in Figure 1C are too small to read. Please adjust the font type and size so that they are more legible.
4. Addition of one or two labels for protein residues in figures with electron density (see Figures 2C, 3A, and 4A & B) would help to orient the general reader.

Point-by-Point Response to Reviewers

Reviewers' comments:

Reviewer #1 (Remarks to the Author):

The manuscript entitled “An engineered crystallization lattice reveals molecular features allowing for potent small-molecule inhibition of HIV-1 entry” by Lai et al. reports multiple crystal structures with and without small molecule entry inhibitors binding to a gp120 binding pocket that show improvements over previous in class candidates. This study extends previous results on the same class of entry inhibitor small molecules and report improved potency over the previous published results from the same groups. The paper contains interesting and important data that would be useful for structure-based drug design. However, the manuscript requires substantial revision to more adequately assess the inferences drawn from the experiments and analyses.

Response

We are grateful to this reviewer for his/her helpful comments. We have now assessed BMS-818251 on a 208-strain cross-clade panel and show it to be a bit more than 10-fold more potent than temsavir on this larger panel (10.6-fold higher geometric mean IC_{50} and 18-fold higher median IC_{50}). Also, we have now assessed binding affinity – and we do see the improved potency reflected in a comparable increase in binding affinity. We have also worked to correct logic in title and elsewhere and appreciate this reviewer pointing out these details.

Main Comments

General: Was the 20-fold improvement in potency also accompanied by an observed improvement in binding for the two identified compounds? There are no binding data shown for the small molecules being discussed and it would be useful to perform binding experiments against a few representative isolates from each of the different clades for reference, preferably from those on the virus panel to assess trends between potency and binding and if clade-specific behavior is apparent. The engineering of lattice contacts is of some importance for this HIV-1 envelope protein as it has been quite difficult to generate high resolution data on Env complexes with antibodies or small molecules. So this in itself is a great achievement. However, in my opinion, the language is a little loose in the title and elsewhere about the lattice engineering where it is the structural information derived from crystals with the engineered lattice that is providing new insights not the lattice engineering per se.

We now have binding data assessed by isothermal calorimetry showing that BMS-818251 has an affinity of 0.047 μ M with BG505 DS-SOSIP, which is ~9-fold higher than the affinity for BMS-626529 (temsavir) under the same condition (0.402 μ M). This is in line with the higher neutralization potency of BMS-818251 against the BG505 virus. It is of interest to test the binding in different strains; however, only limited strains of HIV-1 Env can currently be produced as soluble trimer required for binding analysis, with BG505 being the most well-studied strain. Another strain that can be produced as soluble trimer is B41 with the so called SOSIP mutations. However, this strain showed very high sensitivity to most of the inhibitors in the temsavir family (Figure 1) and thus its binding data will not be as informative as those of the BG505 virus. The ITC binding data is included in the revised manuscript as Supplementary Figure 2 and is cited in the main text associated with main Figure 1.

Line 6-7. The title is misleading as the lattice does not allow for potent small molecule inhibition of HIV-1 entry nor do molecular features *per se*.

We agree with the reviewer that the engineering of the lattice *per se* does not provide the molecular features of potent small molecule inhibition. However, it is important to note that the higher resolution crystal structures required to determine the small molecule interactions would not have been possible without the lattice engineering. The title has been changed to “Lattice engineering enables structural definition of molecular features for potent small-molecule inhibition of HIV-1 entry”. The text has also been modified accordingly.

Line 37. “define accurately”. Do you mean at higher resolution??

Yes, we did mean higher resolution, and have modified line 37 sentence with the reviewer’s suggestion.

Lines 40-41. “and provided a structural explanation for the improved neutralization of BMS-818251” The engineered lattice did not strictly provide this explanation. Noted and see response above.

Line 43. “Crystal lattice engineering thus provide insights” Sorry to keep being picky here but the lattice engineering did not provide these insights, the structures obtained from the crystals derived the crystal lattice engineering provided these. See response above.

Line 66. Please replace “then altered” with “then be altered”. Replaced.

Lines 90-94 and 116-118 mention that BMS-814508 has a 4-fold improvement in potency compared to BMS-626529. Is this true after testing against this minimal virus panel or is it the same as BMS-626529? Lines 90-94 and Line 116-118 seem to have conflicting statements. Is the mean inhibition much lower on a 208-virus panel?

The initial screening showed that BMS-814508 is 4-fold more potent than BMS-626529 against the laboratory-adapted NL4-3 strain. In the subsequent stage of 30-virus verification, BMS-814508 showed only roughly the same potency as BMS-626529. The 30-virus panel is representative of circulating HIV-1 strains and thus the neutralization results on this 30-virus panel is a better indicator for the true potency of small molecule inhibitors than the NL4-3 result. We concluded that BMS-814508 is not a promising hit for further development, and we therefore did not evaluate the BMS-814508 in the costly 208-virus panel assay.

Line 91. Please replace “improve entry” with “improved entry”. Replaced.

Line 91-92. So is the HIV-1 NL4-3 strain an easily neutralized tier1 virus?

Yes, NL4-3 is a commonly used T-cell-line-adapted strain that is easily neutralized by many neutralizing antibodies and entry inhibitors (sometimes referred to as tier 1A strain). The value of using NL4-3 in the initial screening is that it is very stable in cell culture and the screening results are very reproducible. However, to get a realistic appraisal of neutralization potency, hits from initial screening need to be verify in a larger cross-clade virus panel, as we stated in Line 97 (which is now modified for clarity).

Line 99. This is a major experiment on which the punchline of the paper is based. However, it is unclear why only 30 strains were tested instead of the 208-virus panel to infer potency improvements over that reported by Pancera et al. in Nat Chem Biol. To clearly categorize these new small molecules as better leads in the category of gp120 inhibitors, the comparisons really should be head to head, and it is recommended that the compounds be screened against the

identical 208-virus panel reported in the previous study, to remove all doubts of strain-specific improvements, unless a some further rationale is given here of why only the 30-strain panel was used.

Neutralization assays on the 208-virus panel is a resource intensive experiment, hence only the most promising compounds were tested in this format. We did observe that the 30-virus panel is in general in very good agreement with the neutralization results obtained in 208-virus panel. For example, BMS-626529 has a neutralization IC_{50} of ~ 0.04 μM on the 30-virus panel versus an IC_{50} of 0.016 μM on the 208-virus panel.

We are delighted to report that BMS-818251, the most promising hit identified in this study, has now been tested against the 208-virus panel. We determined that the geometric mean IC_{50} of BMS-818251 against the 208-virus panel to be 0.0015 μM , which is in good agreement with the geometric mean IC_{50} of ~ 0.002 μM observed in the 30-virus panel. The 208-virus neutralization data is now included in the manuscript as supplementary table 2 and discussed in the main text associated with main Figure 1.

Line 103. What was the basis of the engineered modifications in the compounds that were designed to aid crystal structure determination?

BMS-386150 has a bromine (“Br” highlighted in red in Figure 1a) modification. Bromine is a much heavier atom than carbon, nitrogen and oxygens and gives a higher intensity in electron density maps and can guide (disambiguate) the model building process. We modified the sentence in Line 103 to explicitly point out that bromine is the modification.

Line 107. Please replace “componenet” with “component”. Replaced.

Line 109. Neutralization of the 30-strain panel generally recapitulated prior observations. What was the rationale of including these 30 HIV-1 strains on this panel? Were these the most difficult to neutralize isolates in the study with BMS-626529 that are now being neutralized by the small molecules in this study?

The rationale behind the 30-virus panel is to provide an estimation of the neutralization IC_{50} on the 208-virus panel by selecting representative strains from initial screening. We assembled the 30-virus panel by choosing a few viruses from each clade spanning the complete range of sensitivity to BMS-626529. As the new 208-virus-panel data of BMS-818251 show, the 30-virus panel indeed is very representative and reliable for initial testing of the neutralization potency of inhibitors in the BMS-626529 family.

Lines 114-115. The modification mentioned on Line 103 to aid structure determination reduces the neutralization potency of BMS-386150. The authors should elaborate on the structural basis for this observation from the Env complex structure at 2.93 Å resolution.

The bromine modification was introduced to aid the model building process in crystal structure determination, with associated cost of reduced potency. From a structure point of view, the methoxy functional group replaced by the bromine was observed to contribute to the binding through hydrophobic interactions with I424, M426 and M434, while the bromine substitution cannot provide such favorable interactions. We’ve added one sentence in the Discussion section to point out the structural basis of the reduced potency of BMS-386150.

Line 120. Please delete “higher”.

We believe “higher” is needed in this sentence and modified the sentence to read “... more than 20-fold higher neutralization potency ...”

Lines 138-140. Is this replicable across isolates from most clades autologous Envs to demonstrate generalizability rather than confined to Envs from strains such as BG505 and B41 that are already known to be favorable for crystallization.

In addition to the BG505 and B41 structures reported here, we have used this lattice to determine structures of several strains that were never crystallized before, including a consensus clade C Env, C97ZA and CAP256 (manuscripts in preparation).

Lines 141-142. “context, mainly due to the lack of structural support of antibody constant region.” I am not sure I fully understand this as the elbow angle is quite flexible and the constant regions are often not well resolved in crystal structures. So where does the ‘lack of structural support’ come from?

Although it is true that elbow region is flexible, it is sometimes observed in crystal structures that the constant region 1 (downstream of elbow sequence) can come into contact with Fv region. In the crystal structure of unliganded 35O22 Fab (4TOY.pdb), both Fv and constant region 1 are well resolved and it is clear from the structure that constant region 1 provided structural support to the Fv in 35O22.

Line 145. Please replace “28.3 Å²” with “28 Å²”. Replaced.

Line 146. Please replace “85.6 Å²” with “86 Å²”. Replaced.

Lines 146-148. Can this also be a stabilization effect of these new small-molecules studied in this work compared to the 484 and BMS-626529 molecules? The claim to generalizability could be supported with a similar resolution structure without the small molecules bound to show the direct effect of lattice engineering with altered chaperones, which is not presently available.

In Figure 2, we compared structures in complex with BMS-378806 before and after using the engineered lattice. It was clear from the comparison that the only difference was the altered chaperones, thus the lower B-factor is a direct consequence of the altered chaperones, not the small molecules. A similar resolution structure could have provided a basis for fair comparison, but not in any sense better than the comparison we provided here.

Line 147. Please replace “71.2 Å²” with “71 Å²”. Replaced.

Line 147. Please replace “201.5 Å²” with “201 Å²”. Replaced.

Line 158. “the unique diffraction number” Is this the number of unique reflections measured? Yes, “unique reflections” is now used in this sentence.

Line 160. Please replace “the information” with “the effective information”. Replaced.

Line 160-161. “constraints obtained from the new crystal more than doubled”. I am not sure what is meant by the constraints doubling?

“Constraints” has been replaced by “data-to-parameter ratio” in this sentence to clarify the meaning.

Line 151. This is really interesting but where are these two glycosylation sites on the 35O22 variants?

One glycan was introduced by mutating I68N to form ⁶⁸NMT⁷⁰ glycan sequon and the second glycan was introduced to the asparagine at the 82B position of heavy chain by introducing a K83T mutation to form a glycan sequon ^{82B}NLT⁸³. These mutations were now specified in the Fig S3.

Line 152. This figure does not mention where the two artificial PNG sites were introduced to improve the yield 10x. Figure S2b simply repeats that 2 PNGs have been introduced. The positions of the two artificial PNG sites are now specified in the legend of Figure S2b (now Figure 3b).

Line 156. Since most known crystal structures of HIV-1 suffer from anisotropy, showing a small section of the structure does not represent the full-picture. The authors should mention if the disordered regions, for e.g., the HR1N etc. is more ordered in the improved lattice. Also, omit maps should be provided for an unbiased evaluation of the improvement due to lattice engineering. The figures do not mention what the illustrated maps are- 2Fo-Fc or omit maps? These need to be clearly labeled in the fig legends. Also, which region is being shown in Fig 2c? There are no labels and no reference to comprehend the orientation.

We've added one sentence toward the end of this paragraph to clarify that intrinsically dynamic regions of Env (such as HR1N) were not resolved in the crystal structures. We believe to resolve those regions, protein engineering on the Env will need to be performed (in contrast to the lattice engineering focus of this manuscript). We've now added a Supplementary Figure 10 to show Fo-Fc omit maps as unbiased evaluation. All the map shown in the main figures are 2Fo-Fc maps and are now clearly labeled in the figure legends. We've labeled Figure 2C with select amino acid positions to orient readers.

Lines 163-168. Are these two regions being contacted by the small molecules? Or this improved resolution can unambiguously be mapped solely to the improvement in lattice contacts by engineering the antibodies?

These two regions are not in direct contact with small molecules, hence the lattice engineering is likely the sole contribution to the improved resolution. We've included the fact that the two regions are not in direct contact with small molecules in the revision.

Line 176. It is unclear how the Env complex with 484 adds to the overall scientific impact of the manuscript.

The compound 484 is currently being actively developed as a probe and an inhibitor for the HIV-1 envelope protein. This compound was first published in *Nature Communications* in 2017 (8:1049) and we provided a docking model in that initial publication. Including the 484 crystal structure in the current manuscript highlights the utility of the improved lattice for determining structures of a promising compound with lower affinity. Structure-based drug optimization can now be reliably performed based on the accurate crystal structure, which was not possible using the previously unmodified lattice. Furthermore, we feel that the inclusion of the 484 crystal structure is appropriate for the current publication as it provides critical follow-up information to the initial publication in *Nature Communications*.

Line 180. Where is the crystallographic table for this co-crystal structure? This should be provided.

The crystallographic table was provided in the original submission, the title was labeled "Compound 484" in the combined crystallographic table.

Line 184. "clear electron density". It is not clear what electron density map is shown in Fig. 3a. Please show the difference Fo-Fc map after molecular replacement or omit map for molecule 484. Also, the protein should be depicted in cartoon/ribbon in a single color and the small molecule shown in a separate figure panel for clear comparison.

The electron density map is 2Fo-Fc and is now labeled in the figure legend. A simulated annealing, difference Fo-Fc map of 484 is now provided in Supplementary Figure 10 to serve as

unbiased evaluation of the final model. The proteins were colored as orange and black to indicate that the protein conformation in the docking structure (black) and co-crystal structure (orange) are essentially identical; it was the small molecule 484 that showed significant differences. We now provide a separate panel in Figure 3a of close-up of compound 484 for clear comparison with the docking conformation.

Line 185. It would be nice to have zoomed figure showing superimposition of docked and crystal structure conformation of compound 484. Also, show zoomed omit map for compound 484. Zoomed figure showing superimposition of docked and crystal structures of compound 484 is now provided in Figure 3a. Zoomed omit maps of all compounds in our crystal structures are now provided in Supplementary Figure 10.

Line 190 and throughout. IC₅₀ should be IC₅₀. We've replaced IC₅₀ with IC₅₀

Lines 200-201. "whether the engineered lattice could provide a structure-function explanation" Sorry, but again the engineered lattice doesn't give that information but the diffraction data acquired from the crystals with the engineered lattice.

We agree and have changed the sentence to read "whether crystal structures derived from the engineered lattice could provide a structure-function explanation"

Line 213. Is this BMS-814251 or BMS-818251? We've corrected this to be BMS-818251 and BMS-814508.

Line 214. The authors infer that the flexibility and therefore lack of contacts with the tail region of BMS-814508 relates to its reduced neutralization potency compared to BMS-818251. What the authors have not commented upon in relating structure-to-function is which contacts in BMS-818251 causes the 25-fold increase in potency compared to BMS-814508 and which contacts are responsible for the cytotoxicity observed at high concentrations of BMS-814508?

We have now specified the functional groups on the BMS-818251 tail that made specific intersections with the envelope protein in the revised text. Because there are multiple interactions between the BMS-818251 tail and the envelope protein, it requires chemical synthesis of a panel of derivatives to knock out the functional groups individually to verify their degree of stabilization effect. This effort is critical for the further optimization of BMS-818251 but is beyond the scope of the current study. It is currently unclear what mechanism led to the observed cytotoxicity of BMS-814508. It is not possible to deduce the contacts of BMS-814508 that led to the off-target toxicity without the knowledge of the mechanism.

Line 218. Please replace "twist" with "twisted". Replaced.

Line 218. Is this piperidine or piperazine ring? Does BMS-818251 have a piperazine or piperadine (Fig. S1)? BMS-818251 has a piperidine ring. This is corrected.

Line 218-219. The cause of these geometric constraints in conformation need more explanation? Perhaps a search for this conformation in the Cambridge crystallographic database would be appropriate to check the conformation?

All available 4-alkenyl piperidines structures (in isolation, not complexed with proteins) in the Cambridge dataset were in chair conformation. However, we note that this higher energy conformation has been observed to spontaneously transition back and forth from the low-energy chair-conformation in molecular dynamics simulations of BMS-626529 (David R. Langley, unpublished data), which adopted a chair conformation in its crystal structure.

Line 220. Which are the interactions in the small molecule core or tail that cause the huge difference in potency between these two molecules?

We now specified the specific interactions that are different in the complex structures of these two molecules. It is likely that the terminal hydroxyl and the amide nitrogen on the tail of BMS-818251 both contributed to the increased potency. However, we note that it requires a large-scale chemical synthesis effort to make the conclusion if both functional groups contributed equally, or if one outweighs the effect of the other. This topic is of interest for further improvement of BMS-818251 but should be addressed in a separate study.

Line 222-223. Overall, it is hard to make any conclusion out of these Env residues in relation to improvement in potency.

This section represents our effort to identify if any specific combination at positions 429 and 432 can lead to potential resistance to BMS-818251. It was to our surprise that all viruses (excluding the three viruses that were neutralized beyond the limit of detection), with a high diversity at positions 429 and 432, showed higher sensitivity to BMS-818251 compared to BMS-814508. Although we cannot exclude the possibility that certain combinations at residues 429 and 432 that are not covered in the 30-virus panel can lead to resistance to BMS-818251, it is important to provide our observation that “improved potency could generally be achieved with variations at positions 429 and 432 that directly interacted with the BMS-818251 tail functional groups” (as we concluded in this section).

Line 224. This should also if possible be tested against the identical 208-virus panel to support this inference.

The identical 208-virus panel has now been tested for BMS-818251 and the conclusion remains valid. We now include the 208-virus data as Supplementary Table 1. BMS-814508 was not tested in the costly 208-virus panel due to its low potency in the 30-virus panel. Thus, the two compounds can only be compared head-to-head based on the 30-virus panel. However, as we noted, the 30-virus panel is very reliable for testing inhibitors in the temsavir family and the comparison based on the 30-virus panel should provide a representative cross-clade evaluation.

Line 229. Why not L116? From figure 4a, it seems L116 is also interacting with the tail of BMS-818251.

L116 is in contact with the thiazole ring where the tail motif extends from but not to the tail itself. L116 is now labeled in the electron density (in the panel on the right) and should be clear to the readers.

Line 232. What is the residue conservation score at positions 117, 429 and 432 of the Env across all recorded unique isolates?

Based on the 2017 LANL sequence collection (6635 sequences), the three positions have the following amino acid % occurrence:

117: K (96.4%); Q (2.3%); others (1.2%)

429: E (46.6%); G (21.9%); R (19.2%); K (7.2%); Q (2.5%); others (2.3%)

432: K (39.4%); R (30.0%); Q (28.6%); others (1.7%)

Line 253. It might have been mentioned elsewhere, but it would be useful to put results in to context: Were only 3 clades - A, B and C tested? If yes, then what were the particular reasons to be limited to these 3 clades?

Diverse strains were studied in many previous reports on the inhibitors of the BMS-626529 family. The clades A, B and C were the most well-studied among all clades. We modified the sentence to clarify that we limited to these three clades because they are well characterized (they also constitute the majority of circulating viruses).

Line 259. Is this the value for only one out of the three gp140 component of the trimer or is this value for the entire trimer?

This value was calculated in the context of the entire trimer, as indicated by “Env-trimer surface” in Line 259.

Line 260. “Surface differences were also present in the epitopes of the crystallization chaperones”. Please describe the differences. Was it only sequenced-based? Or are there corresponding noticeable local conformational differences?

These differences were mostly sequence-based. Only side-chains were different and no local backbone differences were observed.

Line 263. Fig 5b. Is this with or without the small molecules?

This was without the small molecule as we were making a comparison between apo-B41 and apo-BG505 structures.

Line 267. “differences in the three major binding fragments ...”. Please describe these in more detail in the manuscript.

The three major binding fragments referred to the cyan, magenta and green segments in Fig. 5c and as defined in Fig. 3b. This is now added to the manuscript.

Line 272. “consequence of various degrees of conformational pliability...” Does the comparison with the previously published B41 (Barnes et al., Nat Commun, 2018) and BG505 structures, without small molecules bound, justify this hypothesis of local conformational change in this region? Or is this only a consequence of small-molecule binding?

We note that the B41 structure reported by Barnes et al was at a low resolution of 4.95Å and the resulting electron density was very featureless (6CH9.pdb). We provided an apo crystal structure of B41 at an effective resolution of 3.82Å (see crystallographic table), which is substantially higher than the B41 structure reported by Barnes *et al*. We used our apo crystal structure of B41 (which has better defined electron density map than the one obtained by Barnes *et al*) to compare with previous reported BG505 structure at similar resolution (Fig. 5d, lower panel) and concluded that local conformation differences were present in B41 when compared to apo BG505 structure.

Line 276. Please replace “dictat” with “dictate”. Replaced.

Line 292. “while increasing its capacity to facilitate crystallization” Did this result in more crystals from different conditions or solely from an increase in resolution?

More crystallization hits were observed for the improved lattice, in addition to the increased resolution. However, we note that, although crystallized in different conditions, the resulting crystals are the same space group with similar unit cell parameters. The only exception is the identification of a new crystal form (R32 space group) as reported in supplementary figure 9.

Lines 293-294. “engineered lattice yielded twice as many unique reflections as the original crystal, while maintaining similar unit cell parameters”. Sorry but this seems to be due to the increase in resolution if the lattice parameters are effectively identical.

Because the unit cell parameters are essentially the same, the number of unique reflections is proportional to the resolution improvement. It would not have been possible to draw the same conclusion if the improved lattice led to very different unit cell parameters. This sentence modified for clarity.

Lines 297-298. “but has not to our knowledge been used for macromolecular lattice improvement.” Have lattice contacts not been engineered previously in protein crystals? I would have thought they have. So how does this approach differ from those?

The engineering of lattice contacts has certainly been attempted; however, most (if not all) reported cases led to significant change in unit cell parameters or even change in space group, making it very difficult to compare with the original lattice to deduce the effect of lattice engineering. Our improved lattice showed essentially the same unit cell parameter with the original lattice; in this sense, it is conceptually similar to the retro-engineering strategy cited.

Lines 311-312. “trimers adopt slightly different conformations” This contrasts with Line 273 where the authors infer substantial difference in conformation. Which is correct? This should be rephrased to something similar to “the binding pocket for this set of small molecule entry inhibitors around β 20- β 21 hairpin loop region”.

Line 273 has been modified to make it consistent with discussion. Line 311-312 was also modified as suggested.

Lines 327-328. What is the experimental basis of this inference?

This statement was inferred based on comparing the chemical formulas of the two compounds and the potential difference in hydrogen bonding capacity. As part of the discussion, we felt it is important to point out alternative molecular mechanisms that can lead to the higher potency of BMS-818251.

Line 340. “superior neutralization potency” This should only be claimed/validated if the small molecules are screened over an identical 208-virus panel tested for BMS-626529.

We are delighted to report that BMS-818251 is 18-fold more potent than BMS-626529 against the identical 208-virus panel (median IC₅₀ 0.0005 versus 0.009 μ M).

Lines 361 and 382. “37 γ ” “4 γ ” Please correct to the correct unit of temperature throughout the text.

The unit was correct in the submitted Word document and we believe this arose during the conversion to pdf. We will make sure the final proof has a correct unit.

Line 386. It is unclear if the 35O22scFv was deglycosylated after expression and affinity purification and prior to crystallization screens to remove the expression-enhancing artificially-introduced PNGs. The authors should clarify in the methods where these PNGs were introduced and how the scFv was treated post-expression.

We have now clarified where the artificial PNGs were introduced in Supplementary Figure 3. 35O22_scFv was not deglycosylated prior to crystallization. We’ve added this to the Methods.

Line 387. Which variants were used for complex formation out of the four in Supplementary Figure 2b? “1:3:3 molar ratio (gp140 protomer:Fab:scFv) and incubated overnight at room temperature” Was there no slight molar excess of any of the antibodies?

Number 4, the best construct in terms of stability and production yield, was used in crystallization. This is now stated in the Method. There was indeed a slight molar excess of the antibody fragments. We’ve modified it to reflect this fact.

Line 389. What about carbohydrates of Env? Were complexes treated with glycosidases or not? Which 35O22 variants were used, with or without glycosylated variants?

Glycosidase treatment was not performed (now specified in the Methods) and the 35O22 variant used is also specified.

Line 393. Please replace “Data is” with “Data were then”. Replaced.

Line 402. The crystallography table does not mention clearly the following:
What are the components of the complex and how many copies of each component is present in the complex.

The components of the complex are now clearly labeled in the header of the table. The asymmetric unit of each crystal structures contains one copy of gp140 protomer, one copy of 35O22 scFv and one copy of 3H209L Fab, with one small molecule inhibitor occupying the gp120 binding pocket (except the B41 apo structure). This information was added to the Methods.

Line 430. “Journal of medicinal chemistry (2017)” Incomplete reference. Journal abbreviated and missing information added.

Line 458. “Nat Chem Biol (2017)”. Incomplete reference. Missing information added.

Lines 482-483. “Proceedings of the National Academy of Sciences of the United States of America” Please abbreviate journal reference. Journal abbreviated.

Figure 2. A side-by-side comparison of the complexes with Fabs and scFvs would be helpful. This was provided in Supplementary Figure 3.

Figure 2C. Please replace “85.6 Å²” with “86 Å²”. Integers are sufficient for Wilson B and other B factors. Modified as suggested.

Figure 2, Legend, line 4. Please replace “P63” with “P6₃” Modified as suggested.

Figure 3, Legend, line 1. “Accurate”. I do not think accurate is the right word. “Accurate” was removed.

Figure 4, Legend, line 1. “Improved lattice” The structure reveals these features not the improved lattice. We agree and changed it to read “Crystal structures derived from improved lattice”.

Figure 4, Legend, line 7-8. “electron density map” What type of map - 2Fo-Fc?? Yes, “2Fo-Fc” has been added to the legend.

Figure 5, Legend, lines 6-8. “The backbone conformational differences in the structures of BMS-386150 complexed with BG505 (green) or B41 (cyan) are shown as ribbon diagram.” What are the in the top and bottom panels?? They are different but appears to be the same complexes and the same region?

The legend was updated to reflect that the top panel is the comparison between BMS-386150-complexed structures and the bottom panel shows the comparison between apo-form BG505 and B41.

Figure 3. Please depict the protein and small molecule differently (e.g. cartoon/ribbon and ball-and-stick) throughout the manuscript for clear identification.
It is important to show the interactions between protein side chains and the small molecules. We depicted protein in cartoon/ribbon for those parts that have no direct interactions with the small molecule. When protein side chains made contacts with small molecules, they were depicted as sticks to show their relative positions to the small molecules.

Figure 5d. Could the sequence alignment of this region be included to also show differences or similarity?

The sequence alignment between BG505 and B41 was shown in Supplementary Figure 7. The β 20- β 21 region is highlighted in green shade. We changed the color scheme of BG505 and B41 backbone in Figure 5d to avoid confusion.

Supplementary Text

Table S2. Are these complexes with Env and Abs? If yes, please mention it in header of table. Rpim values are missing for all structures. It would be nice to show omit map for each molecule used in complex.

The header of the crystallographic table is now labeled with the complex information. Rpim is included in the revised table. Omit map for each small molecule inhibitors are included as Supplementary Figure 10.

Figure S3. The B41 complex with the engineered scFv does not retain the lattice shown in Barnes et al. Nat Commun, 2018. How does this technique relate to being generalizable? Also, this study lacks the comparison with the Barnes et al. structure with the Fab components at both epitopes with the B41 structures with scFvs reported in the present study.

Barnes *et al* used BG18 instead of antibodies from the PGT122 family (including the the 3H109L we used in this manuscript); BG18 approaches the Env in a different angle ($\sim 30^\circ$ difference) than PGT122 antibodies, which is the main reason they obtained a different lattice (space group R32). We note that the resolution they obtained was very low at 4.95 Å and thus the structure they provided potentially has many errors/inaccuracies (including the bulge in the alpha-9 helix). To obtain the same lattice we reported here, one will need to use the exact pair of crystallization chaperones we provided in this manuscript. We have used this pair to crystallize other strains than the ones reported here, including a consensus clade C, C97ZA and CAP256 (among others that are currently being crystallized). It is our hope that once our manuscript is accepted and published, this pair of crystallization chaperones can be broadly used to determine HIV-1 Env structures with higher resolution and accuracy. Barnes *et al* provided detailed comparison between BG18 and PGT122 in their fig 3 and it is also clear from their analysis that the lattice maintained by PGT122-35O22 chaperones would not be compatible with BG18.

Why these numbers of ratio of reflections to atoms cited may in some respects be technically correct, it is a little misleading when some atoms have been removed and the lattice remains the same. I am not sure how comfortable I am about the possible interpretations from these numbers made on such an analysis.

The data-to-parameter ratio is an important indicator of how stable a structure refinement can be and is related to the quality of the final model. We achieved higher data-to-parameter ratio by increasing resolution (the data) and at the same time decreasing the parameter by removing the constant region 1 of 35O22 (which accounted for $\sim 17\%$ of parameters to be refined, if present). We believe this is a justified analysis to be included in the manuscript.

Figure S4. "5CEZ". What are the extra residues here in 5CEZ??

In this region, residues 556-567 of the 5CEZ structure have no corresponding density at 1.5σ contour level. Four residues (564-567) shown in this figure are now labeled.

Figure S5. Please label residues causing the bulge and some others around it in the figure for orientation. Are there no differences to be observed in the B41 complexes with the Fabs versus

scFvs? There should probably be some comparison with the Barnes et al. Nat Commun 2018 structure?

We have labeled the figure as suggested. The bulge was caused by a register error starting from residue 648. It is clear now from the figures that K655 (with bulged helix) occupied the position of E654 (with bulge fixed). Residues I641 (before bulge), E648 (the bulge) and E654/K655 (downstream of bulge) are labeled. The B41 structure in the Barnes et al Nat Comms 2018 was determined at 4.95Å and the authors apparently used a structure with bulge as starting model. At a resolution of 4.95Å, it would have been impossible for them to identify the register error and to correct the bulge.

Page 14, Table S2

Rsym or Rmerge (%). "(143)" 143.0?? Replaced with 143.0

Please add number of unique reflections measured. Total and unique reflections measured have been added to the crystallographic table.

Resolution (Å) "2.34" "2.50" This is not the actual resolution of this structure- please use the accepted definition of effective resolution when a large number of reflections are missing from the data.

We have added a row for "effective resolution" in the table to account for the missing reflections. The effective resolution was calculated by the following formula used in (F Garces *et al* and IA Wilson, *Immunity* (2015) 43:1053)

Effective resolution was calculated by $Res_{eff} = (\text{Highest resolution}) \times (\text{completeness})^{-1/3}$

B-factors

Please replace "38.9" with "39". Replaced.

Please replace "43.4" with "43". Replaced.

Protein

Please replace "38.1" with "38". Replaced.

Please replace "42.4" with "42". Replaced.

Ligand/ion

Please replace "55.8" with "56". Replaced.

Please replace "61.2" with "61". Replaced.

Water

Please replace "23.26" with "23". Replaced.

Please replace "31.1" with "31". Replaced.

Need to say what the red values are. Some of the higher resolution numbers do not make sense compared to the lower resolution values.

These are the statistics produced by the UCLA Anisotropy Server after the elliptical truncation. These has been changed to italicized font. It has been explained with the & note in the footnote of table, which is in contrast to the conventional overall completeness >50% and $I/\sigma I > 2$ criteria specified by the * note (in regular font).

Page 15, Table S2

Please add number of unique reflections measured. Unique and total reflections measured are added to the crystallographic table.

I / σ

Please replace “17.1 (2.0); 17.38 (1.2)” with “17.1 (2.0); 17.4 (1.2)”. Replaced.

Resolution (Å)

This is not the actual resolution of this structure. Please use the accepted definition of effective resolution when a large number of reflections are missing from the data.

We have added a row for “effective resolution” in the table to account for the missing reflections. The effective resolution was calculated by the following formula used in (F Garces *et al* and IA Wilson, *Immunity* (2015) 43:1053)

Effective resolution was calculated by \$\text{Res}_{\text{eff}} = (\text{Highest resolution}) \times (\text{completeness})^{-1/3}\$

B-factors

Protein, Ligand/ion, Water

Please covert these to integers with no decimal points. All converted.

Reviewer #2 (Remarks to the Author):

This paper by Lai et al reveals a newly engineered crystallization lattice for HIV Env which allows for studying binding of small molecule HIV entry inhibitors at resolutions where the molecular details are more clearly visible. This engineered lattice framework is novel and the methods described here are of interest not only to the HIV community, but also applicable to other fields as well and should be of broad interest to the general scientific community. The work provided here is of high quality and very thorough. This work provides valuable insights into the molecular details of clade specific inhibition of HIV Env by small molecules.

Response

We were delighted by the positive assessment of the manuscript by this reviewer.

Minor points:

1. The overall completeness is low for many of the crystal structures, particularly for B41+35O22_3T2S+3H109L_MM+BMS-386150 and BMS-386150. Could the authors comment as to why this may be?

This was mainly due to the highly anisotropic diffraction of the P6₃ crystal form. The diffraction limit along the best direction was usually more than 1 Å better than the diffraction limits of the other two directions. To include precious high-resolution reflections in our structure refinement, we used the UCLA anisotropy server to judge the diffraction limits along the three principal directions. Effective resolution is now provided in the crystallographic table to account for the lower completeness arose from the elliptical truncation to give readers a comprehensive overview of the diffraction data sets.

2. There seems to be a typo in the number of protein atoms for B41+35O22_3T2S+3H109L_MM in the supplementary materials, please correct. This is corrected from “0,486” to “9,486”.

3. The labels in Figure 1C are too small to read. Please adjust the font type and size so that they are more legible. This was caused by Powerpoint-to-PDF conversion. We will make sure this is legible.

4. Addition of one or two labels for protein residues in figures with electron density (see Figures 2C, 3A, and 4A &B) would help to orient the general reader. We have labeled several protein residues in Figures 2C, 3A and 4A&B to orient the general readers.

Reviewers' Comments:

Reviewer #1:

Remarks to the Author:

Thanks to the authors for responding to my previous comments. I have no further comments and the manuscript now deserves to be published.

REVIEWERS' COMMENTS:

Reviewer #1 (Remarks to the Author):

Thanks to the authors for responding to my previous comments. I have no further comments and the manuscript now deserves to be published.

We are glad that the reviewer #1 is fully satisfied with our response to the initial comments.